# One for All: Universal Topological Primitive Transfer for Graph Structure Learning

**Yide Qiu**[1,†], **Tong Zhang**[1,†], **Xing Cai**[1], **Hui Yan**[1], **Zhen Cui**[2,∗]
[1]School of Computer Science and Engineering, Nanjing University of Science and Technology
[2]School of Artificial Intelligence, Beijing Normal University
{q115025886, tong.zhang}@njust.edu.cn;
caixing11@163.com; yanhui@njust.edu.cn; zhen.cui@bnu.edu.cn

## Abstract

The non-Euclidean geometry inherent in graph structures fundamentally impedes cross-graph knowledge transfer. Drawing inspiration from texture transfer in computer vision, we pioneer topological primitives as transferable semantic units for graph structural knowledge. To address three critical barriers - the absence of specialized benchmarks, aligned semantic representations, and systematic transfer methodologies - we present **G²SN-Transfer**, a unified framework comprising: (i) TopoGraph-Mapping that transforms non-Euclidean graphs into transferable sequences via topological primitive distribution dictionaries; (ii) **G²SN**, a dual-stream architecture learning text-topology aligned representations through contrastive alignment; and (iii) AdaCross-Transfer, a data-adaptive knowledge transfer mechanism leveraging cross-attention for both full-parameter and parameter-frozen scenarios. Particularly, G²SN is a dual-stream sequence network driven by ordinary differential equations, and our theoretical analysis establishes the convergence guarantee of G²SN. We construct **STA-18**, the first large-scale benchmark with aligned topological primitive-text pairs across 18 diverse graph datasets. Comprehensive evaluations demonstrate that G²SN achieves state-of-the-art performance on four structural learning tasks (average 3.2% F1-score improvement), while our transfer method yields consistent enhancements across 13 downstream tasks (5.2% average gains) including 10 large-scale graph datasets. The datasets and code are available at `https://github.com/Yide-Qiu/UGSKT`.

## 1   Introduction

The inherent non-Euclidean geometry of graph structures presents fundamental challenges in preserving hierarchical relational semantics. Topological information encoding, which captures multi-order node relationships through connectivity patterns, has emerged as a critical component in graph representation learning. This capability has driven significant advances across diverse domains: social network analysis through dynamic interaction modeling Zhao *et al.* (2023), biological system understanding via molecular interaction patterns Muzio *et al.* (2021); Li *et al.* (2021), and recommendation systems leveraging user-item relation graphs Wei *et al.* (2022); Sang *et al.* (2024).

Recent research explores graphlets and motifs as structural building blocks for graph representation. While gl2vec Chen and Koga (2019) employs Subgraph Ratio Profiles (SRP) to encode graphlet distributions for classification tasks, and HONE Grover and others (2019) models higher-order interactions through motif-based embeddings, these approaches face three key limitations: (i) handcrafted pattern selection limits semantic generalization; (ii) isolated subgraph analysis disregards global structural contexts; (iii) absence of explicit alignment between structural patterns and transferable semantics.

---

∗Corresponding author: Zhen Cui (zhen.cui@bnu.edu.cn). †These authors contributed equally to this work.

39th Conference on Neural Information Processing Systems (NeurIPS 2025).

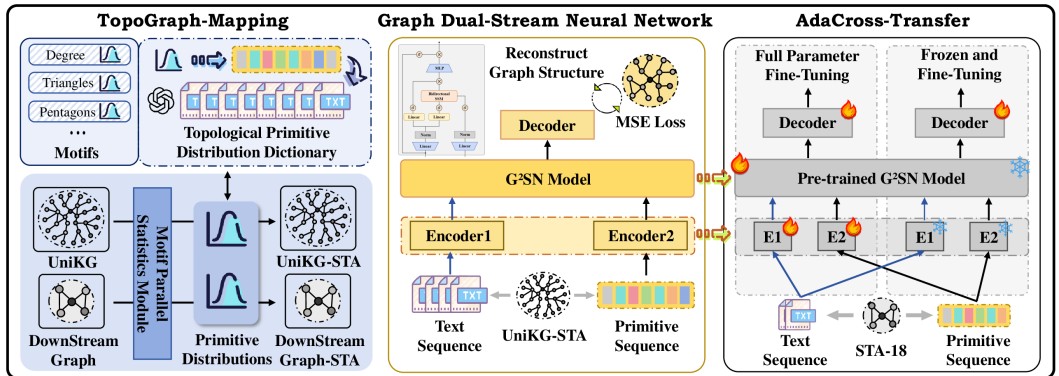

Figure 1: **Overall pipeline of G²SN-Transfer framework**. With topological primitive distribution dictionary-driven, it converts non-Euclidean graph structure into sequential representations, thereby establishing the STA-18 benchmark dataset. Subsequently, the proposed G²SN method undergoes large-scale pre-training on the UniKG-STA to capture transferable structural invariants. The framework culminates in a dual-transfer paradigm encompassing parameter fine-tuning and frozen strategies, enabling adaptive transfer of cross-domain structural graph knowledge.

The MPool framework Islam *et al.* (2023) partially addresses these issues by preserving motif-induced topology in graph pooling, yet fails to establish cross-domain transfer mechanisms. We posit that analogous to transferable texture units in visual neural style transfer, graphs inherently contain topological primitives - statistically recurrent connectivity patterns that encode domain-agnostic structural semantics. Our key insight establishes that the distribution of these primitives forms transferable structural "textures" across graphs, enabling knowledge transfer from structure-rich source graphs to diverse target domains.

While existing methods have advanced graph representation learning, three fundamental limitations persist in structural knowledge transfer: (i) absence of benchmarks for cross-domain structural semantics alignment; (ii) inherent negative transfer risks in aggregation-based approaches due to geometric heterogeneity Pan and Yang (2009); (iii) computational overhead from multi-task auxiliary objectives in current transfer paradigms Xiao *et al.* (2024); Verma and Zhang (2019). These limitations collectively hinder the development of "one-for-all" structural knowledge transfer frameworks Wu *et al.* (2020); Gritsenko *et al.* (2023); Gu *et al.* (2023a,b). Our G²SN-Transfer framework addresses these challenges through three innovations: (i) Topological Primitive Dictionary: We establish a bijective mapping between recurrent connectivity patterns (motifs) and LLM-generated textual descriptors, creating transferable structural-semantic units LLM-Topomotif. (ii) Dual-Modality Sequence Learning: Overcoming the cross-graph semantic gap via contrastive text-topology sequence alignment and structural attention branch for negative transfer mitigation. (iii) Adaptive Knowledge Injection: A parameter-efficient cross-attention mechanism dynamically aligns source-target structural distributions during transfer.

Pre-trained on the largest heterogeneous graph dataset UniKG-STA (UniKG from Qiu *et al.* (2023a)) in the STA-18 benchmark, G²SN-Transfer achieves cross-task structural knowledge transfer across 13 downstream applications through its innovative parameter-efficient cross-attention adapter. This adapter dynamically calibrates structural distribution characteristics between source and target domains, effectively bridging semantic gaps across domains while maintaining base model parameter scale (with fewer than 9% additional parameters). Experimental results demonstrate consistent performance enhancement in diverse scenarios (average improvement of 5.2% across 13 datasets), with particular excellence in ogbl-ppa where it achieves peak improvement of 14.0% accuracy, significantly outperforming conventional transfer methods Li *et al.* (2023).

Our main contributions can be summarized as follows:

i) We establish the first transfer framework defining topological primitives as transferable structural units, formalizing their bijective correspondence with textual semantics through LLM-annotated distribution dictionaries. This framework underpins our STA-18 benchmark – the largest aligned topological-textual graph dataset.

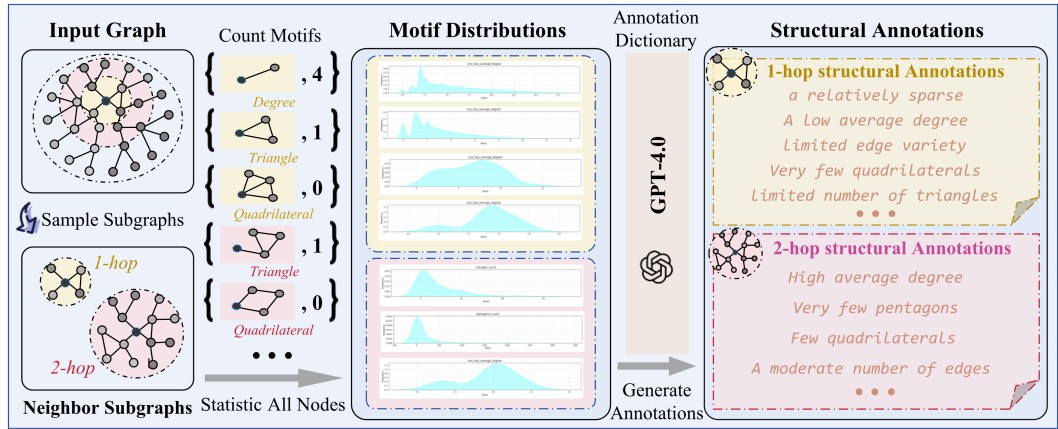

Figure 2: **Annotation generation algorithm**. For any given input graph, the annotation generation algorithm counts the occurrences of various motifs within the one-hop and two-hop subgraphs of each node. Then the motif distribution of the entire graph will be modelled. Based on this quantified distribution, ChatGPT-4.0 translates each component into corresponding structured annotations, producing additional node-level textual annotations.

ii) G²SN, the dual-stream sequence network with contrastive text-topology alignment, is developed to overcome non-Euclidean structural barriers through learnable topological primitives. Our AdaCross-Transfer mechanism enables cross-graph structural knowledge flow via adaptive attention.

iii) We build comprehensive transfer dataset STA-18, the most diverse graph-text benchmark spanning 18 graphs. Systematic evaluations verify our framework's superiority: state-of-the-art performance on 4 structural tasks (3.2% F1 average gain) and consistent improvements across 13 downstream applications (5.2% average gain).

**Overview.** In the remainder of this paper, we first provide the preliminary of State Space Models (SSMs) in Section 2. Then we introduce the graph serialization paradigm TopoGraph-Mapping in Section 3. Subsequently, we present our dual-stream neural network G²SN in Section 4, and our transfer method AdaCross-Transfer in Section 5. Finally, we conduct extensive experiments in Section 6 and conclude our work in Section 7. Notably, we provide related work and preliminaries in the Appendix A and B.

## 2 Preliminaries

SSMs capture the dynamics of systems through hidden state variables, $\mathbf{H}(t) \in \mathbb{R}^N$, which evolve according to an input sequence $x(t) \in \mathbb{R}$. The system dynamics are described by the following linear ordinary differential equations (ODEs):

$$\frac{d\mathbf{H}(t)}{dt} = \mathbf{A}\mathbf{H}(t) + \mathbf{B}x(t), \tag{1}$$

$$\mathbf{Y}(t) = \mathbf{C}\mathbf{H}(t), \tag{2}$$

where $\mathbf{Y}(t)$ is the output, and $\mathbf{A} \in \mathbb{R}^{N \times N}$, $\mathbf{B} \in \mathbb{R}^{N \times 1}$, and $\mathbf{C} \in \mathbb{R}^{1 \times N}$ represent system parameters for state evolution and output projection. To convert the continuous-time model to a discrete form, the Zero-Order Hold (ZOH) method is employed, introducing a timescale $\mathbf{\Delta}$. The discrete-time equivalent of the system is given by:

$$\bar{\mathbf{A}} = \exp(\mathbf{A}\mathbf{\Delta}), \tag{3}$$

$$\bar{\mathbf{B}} = (\bar{\mathbf{A}} - \mathbf{I})\mathbf{A}^{-1}\mathbf{B}\mathbf{\Delta}, \tag{4}$$

leading to the recurrence relations:

$$\mathbf{H}(t) = \bar{\mathbf{A}}\mathbf{H}(t-1) + \bar{\mathbf{B}}x(t), \tag{5}$$

$$\mathbf{Y}(t) = \mathbf{C}\mathbf{H}(t). \tag{6}$$

**Algorithm 1** Structure-Controlled Selection Mechanism

---

**Input**: Sequence $\mathbf{u} \in \mathbb{R}^{V \times L \times D}$; Motif score $\mathbf{m} \in \mathbb{R}^{D}$
**Output**: Sequence $\mathbf{Y} \in \mathbb{R}^{V \times L \times D}$

1: $\mathbf{B} : (V, L, N) \leftarrow \text{Linear}_{\mathbf{B}}(\mathbf{u}, \mathbf{m})$
2: $\mathbf{C} : (V, L, N) \leftarrow \text{Linear}_{\mathbf{C}}(\mathbf{u}, \mathbf{m})$
3: $\boldsymbol{\Delta} : (V, L, D) \leftarrow \log(1 + \exp(\text{Linear}_{\mathbf{A}}(\mathbf{u}, \mathbf{m}) + \text{Parameter}_{\boldsymbol{\Delta}}))$
4: $\mathbf{A}, \bar{\mathbf{B}} : (V, L, D, N) \leftarrow \text{discretize}(\boldsymbol{\Delta}, \text{Parameter}_{\mathbf{A}}, \mathbf{B})$
5: $\mathbf{Y} \leftarrow \text{SSM}(\mathbf{A}, \bar{\mathbf{B}}, \mathbf{C})(\mathbf{u}, \mathbf{m})$
6: **return Y**

---

## 3 TopoGraph-Mapping: Topological Primitive Mapping

We present a graph-to-text framework, as shown in Figure 2, grounded in topological primitive distribution-to-text mapping dictionaries, generating both topological primitive distributions, LLM-Topomotif, and Structural Textual Annotation (STA) for universal graph representation of any given graph data(base). The framework integrates i) an LLM-driven semantic mapping system and ii) a parallel topological primitive quantification module.

Central to our design are **7** fundamental topological primitives characterizing node-centric subgraphs within two hops:

> Average node degree, edge count, edge-type diversity, triangle/quadrilateral/pentagon frequencies, and homogeneous triangle edge probability.

For computational efficiency, we focus on cyclic structures with less than five hops (triangles, quadrilaterals, pentagons) inspired by clique-based graph theory Zhang *et al.* (2024), while incorporating edge heterogeneity and homogeneous cycle likelihoods Blanché *et al.* (2020). Each topological primitive distribution component is bijectively mapped to textual semantics via dictionary $\mathcal{D}$, with annotations ordered by primitive significance. For instance, nodes in the top $k\%$ for average degree receive LLM-generated descriptions like *"This node exhibits exceptionally dense local connectivity"*, validated by human experts to ensure linguistic diversity and statistical fidelity. This dual-channel design enables topological distributions to enhance textual semantics through structural grounding, while textual annotations disambiguate primitive patterns, forming a mutually reinforcing system. The parallel quantification module Wang *et al.* (2016) computes topological primitive distributions across **STA-18** – our benchmark encompassing 18 graph datasets of varied scales. By efficiently extracting and encoding primitive occurrences, it constructs discrete topological distribution profiles, which are subsequently translated into STA codebooks via dictionary $\mathcal{D}$ for cross-graph transfer.

## 4 G²SN: Graph Dual-Stream Neural Network

In Algorithm 1, G²SN has three key parameters ($\mathbf{B}$, $\mathbf{C}$, and $\boldsymbol{\Delta}$) and takes the textual sequence $\mathbf{u}$ and LLM-Topomotif sequence $\mathbf{m}$ as inputs. We first provide a theoretical guarantee for the convergence of G²SN and present a formal derivation of its closed-form solution. Then we theoretically demonstrate that G²SN effectively captures the sequential dependencies for the input sequences $\mathbf{u}$ and $\mathbf{m}$.

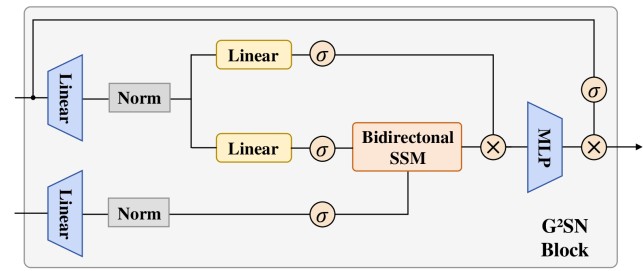

Figure 3: **G²SN** framework. Injecting structural contexts via topological primitive sequence branch.

The discrete zero-order hold (ZOH) method for linear ordinary differential equations (ODEs) provides a recursive formulation, as shown in Eqs. (5) and (6). Leveraging the linear time-invariant (LTI) property,

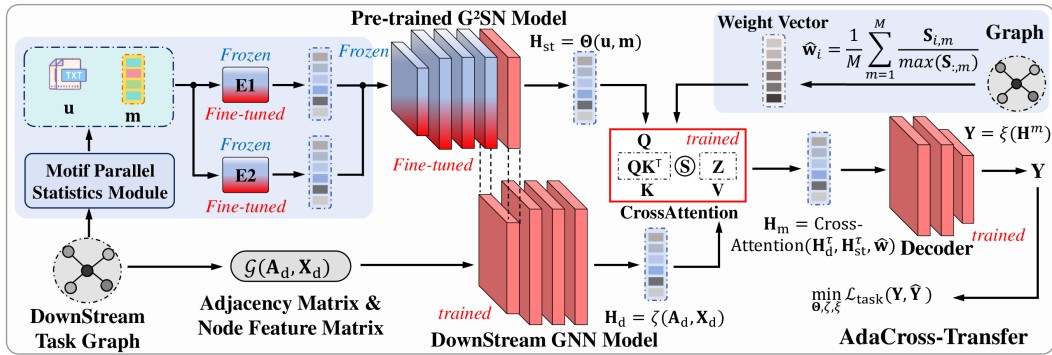

Figure 4: **The overall framework of AdaCross-Transfer**. The *Fine-tuned* represents the "Full Parameter Fine-tuned" scenario, while the *Frozen* indicates the "Frozen and Fine-tuned" scenario.

the recursive expressions for $\mathbf{A}$, $\mathbf{B}$, $\mathbf{C}$, and $\mathbf{\Delta}$ can be parallelized. Furthermore, a global convolutional approach is adopted to compute the output sequence $\mathbf{y} = \mathbf{x} * \bar{\mathbf{K}}$, where the convolution kernel $\bar{\mathbf{K}} \in \mathbb{R}^L$ is defined as:

$$\bar{\mathbf{K}} = [\mathbf{C}\bar{\mathbf{B}}, \mathbf{C}\bar{\mathbf{A}}\bar{\mathbf{B}}, \ldots, \mathbf{C}\bar{\mathbf{A}}^{L-1}\bar{\mathbf{B}}], \tag{7}$$

where $L$ represents the length of the input sequence. Under the ZOH assumption, the closed-form solution for G²SN at time $t$ is given by:

$$\mathbf{H}(t) = \exp(\mathbf{A}t)\mathbf{H}(0) + \int_0^t \exp(\mathbf{A}(t-s))\mathbf{B}x(s)ds. \tag{8}$$

By introducing a time-invariant constant $\gamma$, the closed-form solution for G²SN at time $t+1$ can be expressed as:

$$\mathbf{H}(t+1) = \exp(\mathbf{A})\mathbf{H}(t) + \left( \int_0^1 \exp(\mathbf{A}s)ds \right) \mathbf{B}x(t), \tag{9}$$

which provides a theoretical guarantee for the convergence of G²SN, as stated in Theorem 4.1.

**Theorem 4.1.** *For a given input $x(t)$ from the joint distribution $Q(\mathbf{u}(t), \mathbf{m}(t))$, the G²SN with a generalized form of the time-invariant zero-order hold has a closed-form solution in each discrete sequence step $t$ to $t+1$ and can be computed as $\mathbf{H}(t+1) = \exp(\mathbf{A})\mathbf{H}(t) + (\int_0^1 \exp(\mathbf{A}s)ds)\mathbf{B}x(t)$.*

We propose a structural control mechanism that enables the selection process to depend on both textual sequences and motif quantized distributions. The data-adaptive selection mechanism treats the parameters $\bar{\mathbf{A}}, \bar{\mathbf{B}}, \mathbf{C}$, and $\mathbf{\Delta}$ as functions of the input $x(t)$. This enables the model to dynamically adjust its behavior based on the input data, allowing G²SN to selectively process and retain relevant context, thereby enhancing its capability to handle complex sequential data. To handle additional motif score inputs, the model incorporates the quantized distribution of structural motifs to recall previously encountered textual sequences, improving its ability to reconstruct local structural semantics. In G²SN, the annotation sequence $\mathbf{u}$ and topic score sequence $\mathbf{v}$ jointly serve as inputs.

**Theorem 4.2.** *Under the discrete ZOH assumption, this mechanism enables context-aware information flow through: $\mathbf{Y}(t) = \sum_{q=1}^t e^{(t-q)\mathbf{A}\mathbf{\Delta}}\mathbf{C}(t)\overline{\mathbf{B}}(q)x(q)$ in G²SN, which is jointly governed by topological primitive distributions and textual semantics.*

The model output depends on both the textual annotations and motif sequences, from the Theorem 4.2, with the two jointly controlling the flow of information during propagation. The $\delta(\mathbf{u}, \mathbf{m})\epsilon(\mathbf{u}, \mathbf{m})$ measure the similarity between the current input and prior ones, demonstrating G²SN's ability to capture long-term dependencies. By modifying the forward and backward gradient functions, G²SN, the text encoder $E_1$, and the motif encoder $E_2$ can be jointly trained in an end-to-end manner during pre-training. *Please refer to C.1 for theoretical derivation of Theorem 4.2.*

# 5 AdaCross-Transfer: Graph Structure Knowledge Transfer

**Theorem 5.1.** *G²SN's compositional output* $\mathbf{Y}(t)$, *jointly governed by topological primitive distributions and textual semantics, satisfies Lipschitz-continuous gradient flow, theoretically enabling cross-dataset knowledge transfer through attention reweighting.*

To enable the "one for all" transferring of graph structure knowledge, we design the knowledge transfer method under the "Full Parameter Fine-Tuning" and "Frozen-and-Fine-Tuning" scenarios, as shown in Figure 1 and Figure 4. This method embeds universal and special graph structure semantics as node-level latent features, merging these semantics into the shared representation space, controlled by a structure-aware weight vector $\hat{\mathbf{w}}$. *Please refer to C.3 for the proof of Theorem 5.1.*

Given a downstream task graph dataset $\mathcal{G}(\mathcal{V}, \mathcal{E})$, the transfer process first applies the motif parallel statistics module to produce corresponding structural annotation dataset for each graph dataset. Here, $\mathcal{V}$ and $\mathcal{E}$ represent the set of nodes and edges in $\mathcal{G}$, respectively. Notably, the vocabularies of all dataset are shared, which facilitates the alignment of the generated textual sequence embeddings. The node feature matrix $\mathbf{X}_{\mathrm{d}} \in \mathbb{R}^{n \times d}$ and adjacency matrix $\mathbf{A}_{\mathrm{d}} \in \mathbb{R}^{n \times n}$ of $\mathcal{G}$ are encoded by the downstream task GNN $\zeta$ into the latent variable $\mathbf{H}_{\mathrm{d}} = \zeta(\mathbf{A}_{\mathrm{d}}, \mathbf{X}_{\mathrm{d}})$. We then introduce a transfer encoder $\Theta$ for each downstream task graph, where the first $n$-1 layers are shared from the pre-trained G²SN, and the final layer corresponds to the first layer of the downstream task's GNN $\zeta$. This design ensures that: i) the structural textual latent variable $\mathbf{H}_{\mathrm{st}} \in \mathbb{R}^{n \times d}$ and the node feature latent variable $\mathbf{H}_{\mathrm{d}} \in \mathbb{R}^{n \times d}$ share the same representation space and dimension $d$; and ii) no additional training parameters are introduced, or only fine-tuned parameters are involved.

Considering the uniqueness of each dataset and facilitating the interaction between structural and node features, we compute a data-aware motif attention weight vector $\hat{\mathbf{w}} \in \mathbb{R}^{1 \times n}$ based on the motif quantization distribution. This vector is then integrated into the Cross-Attention module between structural and node features, formalized by:

$$\hat{\mathbf{w}}_i = \frac{1}{M} \sum_{m=1}^{M} \frac{\mathbf{S}_{i,m}}{\max(\mathbf{S}_{:,m})}, \tag{10}$$

where $M$ represents the number of types of motifs, and $\mathbf{S} \in \mathbb{R}^{n \times M}$ is the motif distribution matrix and $\mathbf{S}_{i,m}$ denotes the $m$-th motif score of node $i$. We then interact the two streams at the semantic level through a linear cross-attention layer, which can be formulated as:

$$\mathbf{H}_{\mathrm{st}}^{\tau} = \hat{\mathbf{w}}\mathbf{H}_{\mathrm{st}}, \quad \mathbf{H}_{\mathrm{d}}^{\tau} = (\mathbf{1}_n - \hat{\mathbf{w}})\mathbf{H}_{\mathrm{d}}, \tag{11}$$

$$\text{Linear Attention}(\mathbf{Q}, \mathbf{K}, \mathbf{V}) = \frac{\phi(Q_i)^{\top} \sum_{j=1}^{i} \phi(K_j)V_j^{\top}}{\phi(Q_i)^{\top} \sum_{j=1}^{i} \phi(K_j)}, \tag{12}$$

$$\mathbf{Q}_*, \mathbf{K}_*, \mathbf{V}_* = \mathbf{H}_*^{\tau}\mathbf{W}_Q, \mathbf{H}_*^{\tau}\mathbf{W}_K, \mathbf{H}_*^{\tau}\mathbf{W}_V, \tag{13}$$

$$\mathbf{O}_* = \text{Linear Attention}(\mathbf{Q}_*, \mathbf{K}_*, \mathbf{V}_*), \tag{14}$$

$$\mathbf{H}_*^{\tau} = \text{LayerNorm}(\text{Linear}(\mathbf{O}_* + \mathbf{Q}_*)), \tag{15}$$

where $\mathbf{1}_n$ is an all-ones vector, $* \in \{\mathrm{d}, \mathrm{st}\}$, and $\phi(x) = \text{relu}(x)$ is an activation function. The co-attention operation allows the encoder to emphasize relevant shared semantics while suppressing irrelevant ones. The semantic feature can be concatenated by an MLP, formulated as:

$$\mathbf{H}_{\mathrm{m}} = \text{RELU}(\text{Linear}(\mathbf{H}_{\mathrm{d}}^{\tau}||\mathbf{H}_{\mathrm{st}}^{\tau})), \tag{16}$$

finally, we use a task decoder to return the predicted probability $\hat{\mathbf{Y}}_{\mathrm{task}} = \xi(\mathbf{H}^m)$ as output.

In the "Full Parameter Fine-Tuning" scenario, the networks $E_1$, $E_2$, and $\Theta$ are fine-tuned according to the downstream task. In the "Frozen and Fine-Tuning" scenario, these networks are frozen, and no gradients are computed. Each downstream task adopts a single learning objective, with knowledge transfer targets for node-level, edge-level, and graph-level representation learning tasks, formulated as:

$$\min_{\Theta, \zeta, \xi} \mathcal{L}_{\mathrm{task}} = \begin{cases} \min_{\Theta, \zeta, \xi} \mathcal{L}_{\mathrm{node}}(\mathbf{Y}_{\mathrm{node}}, \hat{\mathbf{Y}}_{\mathrm{node}}), & \text{node-level task}, \\ \min_{\Theta, \zeta, \xi} \mathcal{L}_{\mathrm{edge}}(\mathbf{Y}_{\mathrm{edge}}, \hat{\mathbf{Y}}_{\mathrm{edge}}), & \text{edge-level task}, \\ \min_{\Theta, \zeta, \xi} \mathcal{L}_{\mathrm{graph}}(\mathbf{Y}_{\mathrm{graph}}, \hat{\mathbf{Y}}_{\mathrm{graph}}), & \text{graph-level task}, \end{cases} \tag{17}$$

where $\mathcal{L}_{\text{node}}$, $\mathcal{L}_{\text{edge}}$, and $\mathcal{L}_{\text{graph}}$ represent the loss functions for node-level, edge-level, and graph-level tasks, respectively, and $\hat{\mathbf{Y}}_{\text{node}}$, $\hat{\mathbf{Y}}_{\text{edge}}$, and $\hat{\mathbf{Y}}_{\text{graph}}$ denote the corresponding task labels.

# 6  Experiments

**Overview.**    We conduct extensive experiments to evaluate G²SN's efficacy and universal structural knowledge transfer potential through five key research questions: **Q1**: How does G²SN perform in learning transferable structural representations? **Q2**: Can structural knowledge transfer enhance downstream task performance? **Q3**: How does the cross-attention mechanism facilitate structural knowledge transfer? **Q4**: Is transfer effectiveness correlated with downstream dataset characteristics? **Q5**: What are the computational costs of constructing dual-stream topological primitive-text annotations? Implementation details and hyperparameter configurations are provided in Appendix H.

**Datasets**    Overall, we utilize **36** datasets, including **18** graph datasets and the corresponding **18** STA datasets. For graph datasets, we select **15** large-scale datasets, including **ogbn-arxiv**, **ogbn-products**, **ogbn-proteins**, **ogbn-mag**, **ogbl-ppa**, **ogbl-ddi**, **ogbl-citation2**, **ogbg-molhiv**, **ogbg-molpcba**, and **ogbg-code2** from Hu *et al.* (2020), **Peptides**, **PascalVOC-SP**, **COCO-SP** and **MALNET-TINY** from Behrouz and Hashemi (2024), and **UniKG** from Qiu *et al.* (2023a); as well as **3** small-scale graph datasets: **Cora**, **Pubmed**, and **Citeseer** from Sen *et al.* (2008). Statistics (e.g., graph scale, graph homogeneity, graph density, and token length) of the datasets are summarized in Table 4 and Table 5. The dataset protocols are aligned with those of the OGB benchmark Hu *et al.* (2021) and the original paper Qiu *et al.* (2023a,b); Behrouz and Hashemi (2024).

**Baselines**    We use nine baselines in total. For comparison experiments on **LRGB** (Long Range Graph Benchmark) Dwivedi *et al.* (2022), we utilize **GCN** Kipf and Welling (2017), **GIN** Xu *et al.* (2019), **GatedGCN** Lu *et al.* (2020), **Exphormer** Shirzad *et al.* (2023), **Performer** Choromanski *et al.* (2020), and **BigBird** Zaheer *et al.* (2020) as baseline models. For graph structure transfer, we employ **GCN** Kipf and Welling (2017), **GIN** Xu *et al.* (2019), **GraphSAGE** Hamilton *et al.* (2017), **GraphSAINT** Zeng *et al.* (2020) and **SIGN** Rossi *et al.* (2020) as baseline models. **GCN** and **GIN** are traditional graph convolutional algorithms. We adopt the same sampling techniques as OGB Hu *et al.* (2020) to extend them to large-scale graphs. **GraphSAGE** and **GraphSAINT** aggregate information from neighborhood samples, making them applicable to large-scale graphs. **SIGN** learns the decoupled propagation feature to reduce computational overhead. Note that not every baseline was used on every dataset, as detailed in Table 1 and Table 2.

**Metrics**    We utilize eight evaluation metrics in total: **Acc** (Accuracy), **Rocauc** (Receiver Operating Characteristic Area Under the Curve), **Hits@100**, **Hits@30**, **Mrr** (Mean Reciprocal Rank), **F1 score**, **Ap** (Average Precision) and **MAE** (Mean Absolute Error). These metrics provide a comprehensive evaluation of each baseline model's performance on the test set. Specifically, **Acc** is used for node classification tasks to measure the proportion of correctly predicted samples. **Rocauc** evaluates binary classification models' ability to distinguish between classes. **Hits@100** is used to assess the model's ability to rank the correct target within the top 100 candidates. Similar to **Hits@100**, **Hits@30** evaluates the top 30 candidates. **Mrr** is used to measure the model's performance in ranking tasks by calculating the mean of the reciprocal ranks of the correct answers. **F1 score** provides a harmonic mean of precision and recall in classification tasks. **Ap** is used to assess the model's accuracy in multi-label classification tasks, reflecting the precision across multiple labels. And **MAE** is used to evaluate graph regression tasks.

## 6.1  Graph Structure Learning Experiments on LRGB-STA

**Experimental Setup**    To address **Q1**, we conducted comprehensive graph structure learning comparisons on five tasks from LRGB (Long-Range Graph Benchmark) Dwivedi *et al.* (2022), ensuring evaluation fairness through identical input features and training protocols. Specifically, we focus on comparing several state-of-the-art graph models capable of capturing graph long-rang dependencies, including traditional models such as GCN, GIN, and GatedGCN, as well as various variants within the GraphGPS framework, particularly with respect to the selection of their attention modules. We further isolate structure-controlled selection mechanisms' advantages in structural representation learning by comparing G²SN against dense and sparse variants of Transformers Rampášek *et al.* (2022): Exphormer Shirzad *et al.* (2023) (expander-graph sparsity), Performer Choromanski *et al.* (2020) (kernelized attention), and BigBird Zaheer *et al.* (2020) (block-sparse patterns).

Table 1: **Performance comparison of graph structure learning**. Metrics: AP↑ (higher is better), MAE↓ (lower is better), F1 Score↑ (higher is better), Accuracy↑ (higher is better).

| Model | Peptides-Func AP↑ | Peptides-Struct MAE↓ | PascalVOC-SP F1 Score↑ | COCO-SP F1 Score↑ | MALNET-TINY Accuracy↑ |
|---|---|---|---|---|---|
| GCN | 0.5930±0.0023 | 0.3496±0.0013 | 0.1268±0.0060 | 0.0841±0.0010 | 0.8100±0.0042 |
| GIN | 0.5498±0.0079 | 0.3547±0.0045 | 0.1265±0.0076 | 0.1339±0.0044 | 0.8898±0.0055 |
| GatedGCN | 0.5864±0.0077 | 0.3420±0.0013 | 0.2873±0.0219 | 0.2641±0.0045 | 0.9223±0.0065 |
| GPS+Transformer | 0.6575±0.0049 | 0.2510±0.0015 | 0.3689±0.0131 | 0.3774±0.0150 | OOM (bs=8) |
| GPS+Performer | 0.6475±0.0056 | 0.2558±0.0012 | 0.3724±0.0131 | 0.3761±0.0101 | 0.9264±0.0078 |
| GPS+BigBird | 0.5854±0.0079 | 0.2842±0.0130 | 0.2762±0.0069 | 0.2622±0.0008 | 0.9234±0.0034 |
| Exphormer | 0.6258±0.0092 | 0.2512±0.0025 | 0.3446±0.0064 | 0.3430±0.0108 | 0.9422±0.0024 |
| G²SN (ours) | 0.6824±0.0066 | 0.2462±0.0013 | 0.4199±0.0098 | 0.4022±0.0149 | 0.9391±0.0037 |

Table 2: **Experimental results of structural knowledge transfer**. We use an upward arrow '↑' indicates improved performance. The best and suboptimal performances are highlighted in red and blue, respectively. We report both absolute and relative improvements (Ratio).

| Dataset | Task | Method | Metric | Base | Frozen | Fine-tuned | Ratio (%) |
|---|---|---|---|---|---|---|---|
| ogbn-arxiv | Node Classification | GCN | Accuracy | 0.5238 | 0.5633 (0.0395↑) | 0.5886 (0.0648↑) | 12.37 |
| ogbn-products | Node Classification | SIGN | Accuracy | 0.7423 | 0.7456 (0.0033↑) | 0.7477 (0.0054↑) | 0.73 |
| ogbn-mag | Node Classification | SAGE | Accuracy | 0.3498 | 0.3673 (0.0175↑) | 0.3544 (0.0046↑) | 5.01 |
| Cora | Node Classification | GCN | Accuracy | 0.8110 | 0.8167 (0.0057↑) | 0.8117 (0.0007↑) | 0.70 |
| Pubmed | Node Classification | GCN | Accuracy | 0.7880 | 0.8071 (0.0191↑) | 0.8173 (0.0293↑) | 3.72 |
| Citeseer | Node Classification | GCN | Accuracy | 0.6820 | 0.6873 (0.0053↑) | 0.6981 (0.0161↑) | 2.38 |
| ogbn-proteins | Node Classification | SAGE | ROCAUC | 0.7614 | 0.8175 (0.0561↑) | 0.8076 (0.0462↑) | 7.37 |
| ogbl-molhiv | Link Prediction | GIN | ROCAUC | 0.7761 | 0.7922 (0.0161↑) | 0.7950 (0.0189↑) | 2.44 |
| ogbl-ppa | Link Prediction | SAGE | Hits@100 | 0.1519 | 0.1604 (0.0085↑) | 0.1732 (0.0213↑) | 14.02 |
| ogbl-ddi | Link Prediction | SAGE | Hits@30 | 0.5271 | 0.5549 (0.0278↑) | 0.5601 (0.0330↑) | 6.26 |
| ogbg-citation2 | Graph Classification | SAINT | MRR | 0.8001 | 0.8092 (0.0091↑) | 0.8154 (0.0153↑) | 1.91 |
| ogbg-code2 | Graph Classification | GCN | F1 Score | 0.1515 | 0.1554 (0.0039↑) | 0.1601 (0.0086↑) | 5.68 |
| ogbg-molpcba | Graph Classification | GIN | AP | 0.2744 | 0.2809 (0.0065↑) | 0.2892 (0.0148↑) | 5.39 |

**Experimental Analysis** Table 1 systematically compares G²SN's long-range dependency modeling capabilities across five sequence-intensive tasks. Our framework achieves state-of-the-art performance on four tasks with **2.5%–4.8%** metrics (AP and F1-Score) improvements over sparse attention baselines, demonstrating the critical role of topological primitive distribution-aware structural control. The pre-training robustness on UniKG-STA (Appendix Table 5) further verifies G²SN's dual-stream contrastive alignment effectiveness under distribution shifts. These results confirm that G²SN's structure selected mechanism simultaneously enhances context retention through text-topology interaction modeling while maintaining generalizability across diverse graph prediction scenarios.

## 6.2 Comprehensive Structural Knowledge Transfer Experiments on STA-18

**Experimental Setup** To address **Q2**, we systematically evaluate structural knowledge transfer under both frozen fine-tuning and full-parameter adaptation scenarios. For **Q5**, Table 5 quantifies the computational costs of constructing dual-stream topological primitive-text annotations across STA-18 benchmark datasets. Our transfer framework operates through three coordinated phases: i) Universal Annotation Generation: Each graph dataset is encoded into topology-text pairs using our mapping dictionary. These sequences are vectorized via a shared bag-of-words model to align representation space. ii) Adaptive Encoder Bridging: As illustrated in Fig. 4, the transfer encoder combines the first n-1 layers of pre-trained G²SN with the initial layer of downstream GNNs. This hybrid architecture processes both textual sequences and topological primitive distributions from target domains. iii) Cross-Graph Feature Fusion: Encoder outputs are injected into node-level hidden features through data-adaptive cross-attention modules, which reweight feature importance using structural alignment scores. The fused representations are then processed by task-specific decoders for final predictions. We benchmark transfer effectiveness across both parameter configurations, with implementation details and hyperparameter sensitivity analysis provided in Appendix H.

**Experimental Analysis** From Table 2, we derive three critical insights: i) Universal Performance Enhancement: Structural knowledge transfer delivers consistent gains across all downstream tasks (average +5.2%), confirming its effectiveness in enhancing representation learning for diverse graph data types through topological semantic enrichment. ii) Parameter-Frozen Transfer Superiority: The frozen transfer mode achieves competitive performance with zero training overhead, demonstrating

Table 3: **Ablation studies of transfer methods on six datasets**. **Bold** represents best performance.

| Tranfer Method | Pubmed | ogbn-proteins | ogbl-molhiv | ogbl-ppa | ogbg-citation2 | ogbg-molpcba |
|---|---|---|---|---|---|---|
| Feature Mean | 1.21±0.14 | 2.39±0.12 | 1.18±0.25 | 0.97±0.23 | 0.47±0.28 | 0.22±0.24 |
| Feature Concat | 2.18±0.19 | 2.91±0.17 | 1.37±0.26 | 1.04±0.21 | 1.22±0.24 | 0.76±0.27 |
| Cross-Attention | **2.93±0.11** | **5.61±0.09** | **1.89±0.20** | **2.13±0.14** | **1.53±0.22** | **1.48±0.19** |

that static topological primitive integration alone can provide sufficient inductive bias for improved downstream inference. iii) Fine-Tuning Tradeoff: While full-parameter fine-tuning outperforms frozen transfer in 10 in 13 of tasks via structural encoder optimization, its occasional inferiority (e.g., in ogbn-mag) suggests domain-specific overfitting risks. This highlights the need for future work on balanced optimization strategies that preserve universal topological semantics while adapting to task-specific objectives.

### 6.3 Ablation Study on Transfer Methods

To address **Q3**, we conduct ablation studies on three feature fusion strategies to validate AdaCross-Transfer's adaptive feature routing mechanism. Our experiments across six benchmark datasets (PubMed, ogbn-proteins, ogbl-molhiv, ogbl-ppa, ogbg-citation2, ogbg-molpcba) compare: i) Feature Mean Pooling, ii) Feature Concatenation, and iii) Our Cross-Attention Fusion. As quantified in Table 3 (absolute performance gains) versus Table 2 (relative ratios), the proposed cross-attention method demonstrates universal applicability. AdaCross-Transfer achieves 65.1% average relative improvement over enhancement ratio of conventional fusion methods. These results confirm that cross-attention enables adaptive and thorough interaction between topological primitives and textual semantics - dynamically reweighting feature importance through structural-textual alignment scores. The mechanism particularly excels at resolving semantic ambiguity in molecular graphs (ogbl-molpcba: +94.7%) and protein networks (ogbn-proteins: +92.8%, ogbl-ppa: +104.8%), where conventional methods suffer from feature collision.

### 6.4 What affects the effect of graph structure transfer?

To address **Q4** and establish generalizable principles for effective structural knowledge transfer, we derive three empirical conclusions from the above experiments: i) **Density Correlation**: Transfer efficacy exhibits statistically significant correlations with average degree distributions (pearson correlation coefficient $\rho=0.82$, significance $p<0.01$, the derivations refer to Appendix F). Node classification tasks reveal substantial gains in high-degree graphs: ogbn-arxiv (degree=13.77, +12.37%) and ogbn-proteins (597.00, +7.37%), contrasting with minimal improvement in low-degree Cora (4.01, +0.70%). This demonstrates dense topological primitives in high-degree graphs better preserve transferable structural semantics. ii) **Homogeneity Superiority**: Heterogeneous graphs show reduced transfer efficiency despite comparable degree metrics - ogbn-mag (degree=21.76, +5.01%) underperforms homogeneous counterparts by 2.3-4.8%, suggesting structural heterogeneity introduces semantic fragmentation that impedes cross-graph alignment. iii) **Task-Specific Sensitivity**: Structural transfer achieves superior gains in link prediction (avg. +7.57%) and graph classification (+4.33%) versus node classification (+3.23%), due to their inherent dependency on structural topology rather than feature-driven decisions vulnerable to attribute noise. These findings collectively establish two guidelines: i) Prioritize structural knowledge transfer for topology-intensive tasks (link prediction/graph classification) on degree-dense homogeneous graphs. ii) Develop heterogeneity-aware transfer methods to address structural fragmentation in complex graphs, which implying the interplay between graph heterogeneity and transfer efficiency as a promising direction for subsequent research.

## 7 Conclusion

We present **G²SN-Transfer**, a universal graph structure learning framework that enables cross-domain knowledge transfer through topological primitive-text dual-stream sequence alignment. The framework introduces: (i) **TopoGraph-Mapping** for serializing non-Euclidean graphs into transferable sequences via topological primitive distribution dictionaries, thereby establishing the STA-18 benchmark; (ii) **G²SN**, a dual-stream sequence network achieving text-topology contrastive alignment through adaptive attention routing, with convergence guarantees under discrete zero-order hold assumptions; (iii) **AdaCross-Transfer**, a data-adaptive cross-attention mechanism. Comprehensive evaluations demonstrate state-of-the-art performance on 4 structural learning tasks (3.2% average F1-score gain) and consistent improvements across 13 downstream applications (5.2% average lift), validating the effectiveness of topological primitives as universal structural semantic units.

## Acknowledgement

This work was supported by the National Natural Science Foundation of China (Grants No. 62476133) and the Fundamental Research Funds for the Central Universities (Grant No. 11300-312200502507).

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

# A Related Work

Table 4: **Statistics for the graph datasets**. Where 'N.C.' denotes 'Node Classification task', 'L.P.' represents 'Link Prediction task', 'G.C.' is 'Graph Classification task' and 'G.R.' denotes 'Graph Regression task'.

| Dataset | #Nodes | #Edges | #Avg.Degree | #Heterogeneity | #Task | #Metric | #Domain |
|---|---|---|---|---|---|---|---|
| Cora | 2,708 | 5,429 | 4.01 | No | N.C. | Acc | Citation |
| Citeseer | 3,327 | 4,732 | 2.84 | No | N.C. | Acc | Citation |
| Pubmed | 19,717 | 44,338 | 4.50 | No | N.C. | Acc | Citation |
| ogbn-proteins | 132,534 | 39,561,252 | 597.00 | No | N.C. | ROCAUC | Biology |
| ogbn-arxiv | 169,343 | 1,166,243 | 13.77 | No | N.C. | Acc | Citation |
| ogbn-mag | 1,939,743 | 21,111,007 | 21.76 | Yes | N.C. | Acc | Citation |
| ogbn-products | 2,449,029 | 61,859,140 | 50.52 | No | N.C. | Acc | Product |
| ogbl-ddi | 4,267 | 1,334,889 | 625.68 | No | L.P. | Hits@30 | Biology |
| ogbl-ppa | 576,289 | 30,326,273 | 105.24 | No | L.P. | Hits@100 | Biology |
| ogbl-citation2 | 2,927,963 | 30,561,187 | 20.74 | No | L.P. | Mrr | Citation |
| ogbg-molhiv | 1,048,738 | 1,130,992 | 2.15 | No | G.C. | ROCAUC | Biology |
| ogbg-molpcba | 11,386,154 | 12,305,804 | 2.16 | No | G.C. | AP | Biology |
| ogbg-code2 | 56,683,173 | 56,230,432 | 1.98 | No | G.C. | F1 score | Code |
| Peptides | 2,344,231 | 4,773,905 | 2.04 | No | G.C. & G.R. | AP & MAE | Biology |
| PascalVOC-SP | 5,443,587 | 30,777,727 | 5.65 | No | N.C. | F1 score | CV |
| COCO-SP | 58,795,093 | 332,095,498 | 5.64 | No | N.C. | F1 score | CV |
| MalNet-Tiny | 7,051,500 | 14,299,500 | 2.03 | No | G.C. | Acc | Cybersecurity |
| UniKG | 77,312,474 | 641,738,096 | 16.60 | Yes | N.C. | Acc | Universal |

Table 5: **Statistics for STA-18 benchmark**.

| Dataset | #Nodes | #Tokens | #Lengths | #Spaces | #Generation Time | #Task | #Metric | #Domain |
|---|---|---|---|---|---|---|---|---|
| Cora-STA | 2,708 | 342,249 | 126.4 | 0.11 MB | 7.07 s | N.C. | Acc | Citation |
| Citeseer-STA | 3,327 | 420,580 | 126.4 | 0.13 MB | 4.65 s | N.C. | Acc | Citation |
| Pubmed-STA | 19,717 | 2,495,065 | 126.5 | 0.75 MB | 108.94 s | N.C. | Acc | Citation |
| ogbn-proteins-STA | 132,534 | 17,028,774 | 127.7 | 5.06 MB | 6.83 h | N.C. | ROCAUC | Biology |
| ogbn-arxiv-STA | 169,343 | 21,338,818 | 126.0 | 6.46 MB | 616.12 s | N.C. | Acc | Citation |
| ogbn-mag-STA | 1,939,743 | 242,542,736 | 125.0 | 74.00MB | 4.56 h | N.C. | Acc | Citation |
| ogbn-products-STA | 2,449,029 | 312,639,453 | 127.7 | 93.42MB | 113.14 h | N.C. | Acc | Product |
| ogbl-ddi-STA | 4,267 | 549,291 | 128.7 | 0.16 MB | 845.25 s | L.P. | Hits@30 | Biology |
| ogbl-ppa-STA | 576,289 | 73,407,315 | 127.4 | 21.98 MB | 3.01 h | L.P. | Hits@100 | Biology |
| ogbl-citation2-STA | 2,927,963 | 36,996,424 | 126.4 | 111.69 MB | 12.81 h | L.P. | Mrr | Citation |
| ogbg-molhiv-STA | 1,048,738 | 130,250,852 | 124.1 | 40.02 MB | 398.34 s | G.C. | ROCAUC | Biology |
| ogbg-molpcba-STA | 11,386,154 | 1,528,535,562 | 134.4 | 433.85 MB | 1.17 h | G.C. | AP | Biology |
| ogbg-code2-STA | 56,683,173 | 7,528,675,744 | 132.8 | 2.11 GB | 2.84 h | G.C. | F1 score | Code |
| Peptides-STA | 2,344,231 | 299,346,341 | 127.7 | 89.45 MB | 930.91 s | G.C. & G.R. | AP & MAE | Biology |
| PascalVOC-SP-STA | 5,443,587 | 714,198,614 | 131.2 | 213.60 MB | 0.92 h | N.C. | F1 score | CV |
| COCO-SP-STA | 58,795,093 | 7,619,844,052 | 129.6 | 2.22 GB | 6.58 h | N.C. | F1 score | CV |
| MalNet-Tiny-STA | 7,051,500 | 897,655,950 | 127.3 | 268.23 MB | 0.77 h | G.C. | Acc | Cybersecurity |
| UniKG-STA | 77,312,474 | 10,274,827,794 | 132.9 | 6.08 GB | 214.22 h | N.C. | Acc | Universal |

Graph Structure Learning have gained significant traction for their ability to learn complex graph representations, particularly through the use of structural subcomponents like **Graphlets** and **Motifs**. These subcomponents are small non-isomorphic induced subgraphs, serve as fundamental building blocks for capturing local structural information in graphs. They have been employed in various network embedding techniques to improve the expressiveness of GNNs.

Several studies have explored the utilization of graphlets in structural representation learning. For instance, the gl2vec model generates embeddings by computing the Subgraph Ratio Profile (SRP) of graphlets, which encodes the distribution of different types of graphlets in a network. This approach has demonstrated improvements in classification tasks, particularly when combined with other feature extraction methods. Similarly, the work by Rossi et al. proposed the Higher-Order Network Embedding (HONE) framework, which uses subgraph patterns, including motifs and graphlet orbits, to capture higher-order network structures. Other notable approaches include leveraging graph convolutional networks (GCNs) with motif-based attention mechanisms. For example, MPool, a motif-based graph pooling method, has shown that incorporating motif structures can improve graph classification performance by preserving essential topological information during pooling.

Our work differentiates itself from existing methods by focusing on the distribution of motifs and constructing these distributions as text sequences, akin to a growth series of motifs. This novel approach allows for the capture of both the frequency and evolution of motifs within a graph, which can be beneficial for tasks requiring a more nuanced understanding of network dynamics. Unlike the

gl2vec approach, which relies on random models to generate SRPs, our method does not depend on such models, thereby offering potentially more robust and consistent embeddings **across** different datasets. By integrating this with existing language model frameworks, our approach not only captures the local structure but also the evolution of these structures, offering a comprehensive view of the graph's topology.

## B    More Preliminaries

### B.1    Notation

A graph can be defined as $\mathcal{G} = (\mathcal{V}, \mathcal{E})$, in which $\mathcal{V}$ and $\mathcal{E}$ represent the sets of nodes and edges, respectively. For homogeneous graph, the number of types of nodes and edges is strictly equal to one. For heterogeneous graphs, the sum of the types of nodes and edges is strictly greater than two. Formally, a heterogeneous graph can be defined as follows:

$$\mathcal{G} = (\mathcal{V}, \mathcal{E}, \mathcal{C}, \mathcal{R}, \phi_v, \phi_e),$$

where:

- $\mathcal{V} = \{v_1, v_2, \ldots, v_N\}$ denotes the set of nodes, and $N$ is the total number of nodes.
- $\mathcal{E} = \{e_{ij}\}$ denotes the set of directed edges between nodes.
- $\mathcal{C} = \{c_1, c_2, \ldots, c_L\}$ is the set of node types, and $L$ is the total number of node types.
- $\mathcal{R} = \{r_1, r_2, \ldots, r_M\}$ is the set of edge types, and $M$ is the total number of edge types.
- $\phi_v : \mathcal{V} \to \{c_1, c_2, \ldots, c_L\}$ represents the node type function that maps nodes to their corresponding types.
- $\phi_e : \mathcal{E} \to \{r_1, r_2, \ldots, r_M\}$ represents the edge type function that maps edges to their corresponding types.

### B.2    Graph Neural Network Paradigm

The fundamental idea behind GNNs is to iteratively update the representation of each node by aggregating information from its neighbors. This process can be generally formulated as:

$$\mathbf{h}_v^{(k)} = \text{AGGREGATE}^{(k)} \left( \left\{ \mathbf{h}_u^{(k-1)} : u \in \mathcal{N}(v) \right\} \right), \tag{18}$$

$$\mathbf{h}_v^{(k)} = \text{COMBINE}^{(k)} \left( \mathbf{h}_v^{(k-1)}, \mathbf{h}_v^{(k)} \right), \tag{19}$$

where $\mathbf{h}_v^{(k)}$ denotes the representation of node $v$ at the $k$-th layer, $\mathcal{N}(v)$ represents the set of neighbors of $v$, and AGGREGATE($\cdot$) and COMBINE($\cdot$) are functions that aggregate information from the neighbors and combine it with the node's previous representation, respectively.

### B.3    Frozen/Fine-tuned Knowledge Transfer

Frozen knowledge transfer aims to inject prior semantic knowledge, acquired through pre-training, into downstream tasks without additional training cost. The large-scale training set that provides prior knowledge is referred to as the support set, while the dataset for downstream tasks is termed the target set. This transfer is achieved by pre-training a shared semantic space. Unlike zero-shot learning, frozen knowledge transfer does not impose strict constraints on class labels. In TA-LSSR framework, we use UniKG-STA, a dataset with rich structural information, as the support set, and various downstream graph datasets as the target sets to ensure alignment with the shared semantic space hypothesis. Fine-tuned knowledge transfer aims to optimize both the pre-trained model and the downstream task model through end-to-end training. The objective is to adapt existing knowledge to the downstream task with minimal training data and parameters, while aligning with the task-specific optimization goals. In our fine-tuning scenario, we first pre-train the G²SN model on UniKG-STA dataset, and fine-tune it on various downstream graph datasets.

## C  Proofs for Theorems

### C.1  Proofs of Theorem 1

In this section, we prove that the closed-formed solution of G²SN can be calculated as:

$$\mathbf{H}(t+1) = \exp(\mathbf{A})\mathbf{H}(t) + \left(\int_0^1 \exp(\mathbf{A}s)ds\right)\mathbf{B}x(t). \tag{20}$$

From Eq. 1, the linear time-invariant ODE function of the state space models can be described:

$$\frac{d\mathbf{H}(t)}{dt} = \mathbf{A}\mathbf{H}(t) + \mathbf{B}x(t), \tag{21}$$

where $x(t)$ is the input function of the discrete sequences and $\mathbf{H}(t)$ denotes the hidden state variables. First and foremost, we derive the $\mathbf{H}(k)$ in G²SN. Multiply both sides of the ODE function by $\exp(-\mathbf{A}t)$:

$$\exp(-\mathbf{A}t)\frac{d\mathbf{H}(t)}{dt} = \exp(-\mathbf{A}t)\mathbf{A}\mathbf{H}(t) + \exp(-\mathbf{A}t)\mathbf{B}x(t), \tag{22}$$

according to the partial integration, the above formula can be derived as follows:

$$\frac{d}{dt}(\exp(-\mathbf{A}t)\mathbf{H}(t)) = \exp(-\mathbf{A}t)\mathbf{B}x(t), \tag{23}$$

integrating both sides with respect to $t$:

$$\exp(-\mathbf{A}t)\mathbf{H}(t) - \exp(0)\mathbf{H}(0) = \int_0^t \exp(-\mathbf{A}s)\mathbf{B}x(s)ds, \tag{24}$$

after a merging, we can prove that:

$$\mathbf{H}(t) = \exp(\mathbf{A}t)\mathbf{H}(0) + \int_0^t \exp(\mathbf{A}(t-s))\mathbf{B}x(s)ds. \tag{25}$$

Under the assumption of zero-order hold (ZOH), $x(t)$ is constant during each time interval $\Delta t$, which means $x(t+\gamma) = x(t)$ if $\gamma < \Delta t$. Let $v(s) = t+1-s$, the $\mathbf{H}(t+1)$ can be further derived as:

$$\mathbf{H}(t+1) = \exp(\mathbf{A}(t+1))\mathbf{H}(0) + \int_0^{t+1} \exp(\mathbf{A}(t+1-s))\mathbf{B}x(s)ds \tag{26}$$

$$= \exp(\mathbf{A}t + \mathbf{A})\mathbf{H}(0) + \exp(\mathbf{A})\left(\int_0^t \exp(\mathbf{A}(t-s))\mathbf{B}x(s)ds + \int_0^1 \exp(\mathbf{A}(t-s))\mathbf{B}x(s)ds\right) \tag{27}$$

$$= \exp(\mathbf{A})\left(\exp(\mathbf{A}t)\mathbf{H}(0) + \int_0^t \exp(\mathbf{A}(t-s))\mathbf{B}x(s)ds\right) + \int_0^1 \exp(\mathbf{A}(t+1-s))\mathbf{B}x(s)ds \tag{28}$$

$$= \exp(\mathbf{A})\mathbf{H}(t) - \left(\int_{v(t)}^{v(t+1)} \exp(\mathbf{A}v)dv\right)\mathbf{B}x(t) \tag{29}$$

$$= \exp(\mathbf{A})\mathbf{H}(t) + \left(\int_0^1 \exp(\mathbf{A}s)ds\right)\mathbf{B}x(t). \quad \square \tag{30}$$

### C.2  Proofs of Theorem 2

In this section, we prove that the context dependencies of G²SN are controlled by the structural semantics $\mathbf{u}$ and $\mathbf{m}$. According to Algorithm 1, G²SN has two essential parameters $\bar{\mathbf{B}}$ and $\mathbf{C}$, which are functions of text annotation $\mathbf{u}$ and motif sequence $\mathbf{m}$. Thus, they can be represented as $\delta(\mathbf{u}, \mathbf{m})$ and $\epsilon(\mathbf{u}, \mathbf{m})$, respectively. Under the assumption of ZOH, discrete-time equivalent of ODE can be represented as:

$$\mathbf{H}(t) = \bar{\mathbf{A}}(t)\mathbf{H}(t-1) + \bar{\mathbf{B}}(t)x(t), \tag{31}$$

$$\mathbf{Y}(t) = \mathbf{C}(t)\mathbf{H}(t). \tag{32}$$

Eq. 31 is an iterative equation. The $\mathbf{H}(t)$ can be decomposed as follows:

$$\mathbf{H}(t) = \prod_{i=0}^{t-2} \bar{\mathbf{A}}(t-i)\bar{\mathbf{B}}(1)x(1) + \cdots + \prod_{i=0}^{t-q} \bar{\mathbf{A}}(t-i)\bar{\mathbf{B}}(q-1)x(q-1) + \cdots + \bar{\mathbf{B}}(t)x(t). \tag{33}$$

From the Eq. 3, we have $\bar{\mathbf{A}} = \exp(\mathbf{A}\boldsymbol{\Delta})$, thus the $\mathbf{Y}(t)$ can be represented as:

$$\mathbf{Y}(t) = \mathbf{C}(t)\left(\prod_{i=0}^{t-2} \exp(\mathbf{A}(t-i)\boldsymbol{\Delta})\bar{\mathbf{B}}(1)x(1) + \cdots + \prod_{i=0}^{t-q-1} \exp\left(\mathbf{A}(t-i)\boldsymbol{\Delta}\right)\bar{\mathbf{B}}(q)x(q) + \cdots + \bar{\mathbf{B}}(t)x(t)\right), \tag{34}$$

$$= \mathbf{C}(t)\left(\exp(\sum_{i=0}^{t-2} \mathbf{A}(t-i)\boldsymbol{\Delta})\bar{\mathbf{B}}(1)x(1) + \cdots + \exp(\sum_{i=0}^{t-q-1} \mathbf{A}(t-i)\boldsymbol{\Delta})\bar{\mathbf{B}}(q)x(q) + \cdots + \bar{\mathbf{B}}(t)x(t)\right), \tag{35}$$

$$= \exp(\sum_{i=0}^{t-2} \mathbf{A}(t-i)\boldsymbol{\Delta})\mathbf{C}(t)\bar{\mathbf{B}}(1)x(1) + \cdots + \exp(\sum_{i=0}^{t-q-1} \mathbf{A}(t-i)\boldsymbol{\Delta})\mathbf{C}(t)\bar{\mathbf{B}}(q)x(q) + \cdots + \mathbf{C}(t)\bar{\mathbf{B}}(t)x(t), \tag{36}$$

where the $\mathbf{A}(t)$ are time-invariant parameter matrices, thus we have:

$$\mathbf{Y}(t) = e^{(t-1)\mathbf{A}\boldsymbol{\Delta}}\mathbf{C}(t)\bar{\mathbf{B}}(1)x(1) + \cdots + e^{(t-q)\mathbf{A}\boldsymbol{\Delta}}\mathbf{C}(t)\bar{\mathbf{B}}(q)x(q) + \cdots + \mathbf{C}(t)\bar{\mathbf{B}}(t)x(t) \tag{37}$$

$$= \sum_{q=1}^{t} e^{(t-q)\mathbf{A}\boldsymbol{\Delta}}\mathbf{C}(t)\overline{\mathbf{B}}(q)x(q) \qquad \square \tag{38}$$

The $x(t)$ is $t$-th input of $\mathbf{M}_{\mathbf{u},\mathbf{m}}, i.e., x(t) = \mathbf{M}(t)$, and $\bar{\mathbf{B}}(q)$ is the parameter matrix $\text{Linear}_B(\delta(\mathbf{u}(q), \mathbf{m}(q)))$, and $\mathbf{C}(t)$ is the parameter matrix $\text{Linear}_C(\epsilon(\mathbf{u}(t), \mathbf{m}(t)))$. Thus $\delta(\mathbf{u}(t), \mathbf{m}(t))$ can be considered as one query of $x(t)$, and $\epsilon(\mathbf{u}(q), \mathbf{m}(q))$ can be considered as one key of $x(q)$. Then, $\delta(\mathbf{u}, \mathbf{m})\epsilon(\mathbf{u}, \mathbf{m})$ can measure the similarity between current input to the previous ones and selectively copy the previous input, which demonstrates G²SN's long-term dependencies for text annotations and motif sequences.

### C.3 Proofs of Theorem 3

*Proof.* Let $\mathbf{Y}(t) = \sum_{q=1}^{t} e^{(t-q)\mathbf{A}\boldsymbol{\Delta}}\mathbf{C}(t)\bar{\mathbf{B}}(q)x(q)$. We prove its gradient flow satisfies Lipschitz continuity as follows:

The gradient w.r.t parameters contains:

$$\nabla\mathbf{Y}(t) = \underbrace{\sum_{q=1}^{t} \mathbf{A}\boldsymbol{\Delta}e^{(t-q)\mathbf{A}\boldsymbol{\Delta}}\mathbf{C}\bar{\mathbf{B}}x(q)}_{\text{Time evolution term}} + \underbrace{\sum_{q=1}^{t} e^{(t-q)\mathbf{A}\boldsymbol{\Delta}}\nabla_\theta[\mathbf{C}\bar{\mathbf{B}}]x(q)}_{\text{Parameter dynamics}}, \tag{39}$$

where $\theta$ denotes model parameters.

For matrix exponential operators:

$$\|e^{\mathbf{A}\boldsymbol{\Delta}t_1} - e^{\mathbf{A}\boldsymbol{\Delta}t_2}\|_2 \leq \|\mathbf{A}\boldsymbol{\Delta}\|_2 e^{\|\mathbf{A}\boldsymbol{\Delta}\|_2 \max(t_1, t_2)}|t_1 - t_2|, \tag{40}$$

yielding Lipschitz constant $L_1 = \|\mathbf{A}\boldsymbol{\Delta}\|_2 e^{\|\mathbf{A}\boldsymbol{\Delta}\|_2 T}$ for maximum timestep $T$.

The parameter matrices $\mathbf{C}(t) = \text{Linear}_C(\epsilon(\mathbf{u}, \mathbf{m}))$ and $\bar{\mathbf{B}}(q) = \text{Linear}_B(\delta(\mathbf{u}, \mathbf{m}))$ are constrained by:

i) **Spectral Normalization**:

$$\sigma_{\max}(\mathbf{W}_B), \sigma_{\max}(\mathbf{W}_C) \leq 1, \tag{41}$$

where $\mathbf{W}_B$, $\mathbf{W}_C$ are weight matrices in $\text{Linear}_B$, $\text{Linear}_C$.

ii) **K-Lipschitz Similarity Functions**:

$$\|\delta(\mathbf{u}_1, \mathbf{m}_1) - \delta(\mathbf{u}_2, \mathbf{m}_2)\| \leq K(\|\mathbf{u}_1 - \mathbf{u}_2\| + \|\mathbf{m}_1 - \mathbf{m}_2\|), \tag{42}$$

with analogous bound for $\epsilon(\cdot, \cdot)$.

Combining these yields:

$$\|\nabla_\theta[\mathbf{C}\bar{\mathbf{B}}]\| \leq L_2(\|\mathbf{u} - \mathbf{u}'\| + \|\mathbf{m} - \mathbf{m}'\|), \tag{43}$$

where $L_2 = 2K\sqrt{d}$ with $d$ as hidden dimension.

The total Lipschitz constant becomes:

$$L = L_1 + L_2 \cdot \max_{t,q} \|\mathbf{C}(t)\bar{\mathbf{B}}(q)\|_2. \tag{44}$$

Spectral normalization ensures $\|\mathbf{C}\bar{\mathbf{B}}\|_2 \leq 1$, thus $L$ is finite and data-independent.

**Cross-Dataset Transfer Mechanism**. The Lipschitz continuity guarantees that attention reweighting through $\delta(\cdot), \epsilon(\cdot)$ maintains bounded feature distortion when transferring between domains $\mathcal{D}_S$ (source) and $\mathcal{D}_T$ (target):

$$\|\mathbf{Y}_{\mathcal{D}_S}(t) - \mathbf{Y}_{\mathcal{D}_T}(t)\| \leq L \cdot \text{MMD}(\mathcal{D}_S, \mathcal{D}_T), \tag{45}$$

where MMD denotes Maximum Mean Discrepancy. This theoretically enables stable knowledge transfer through attention adjustment. □

# D Details of Datasets

## D.1 Graph Datasets Statistics

Table 4 presents statistics for the utilized pre-training graph dataset **UniKG** and downstream graph datasets. The table compares characteristics such as the number of nodes and edges, graph density, heterogeneity, task type, evaluation metric, and application domain for each dataset. Specially, "#Nodes" shows the number of nodes for each dataset. Datasets vary significantly in size, from small networks like **Cora** (2,708 nodes) to much larger networks like **ogbn-products** (2,449,029 nodes) and **UniKG** (77,312,474 nodes). "#Edges" reports the number of edges in the graph. For instance, **Pubmed** contains 44,338 edges, while the largest dataset, **UniKG**, has 641,738,096 edges, illustrating the scale variation across datasets. Graph "#Density" is the ratio of actual edges to possible edges in the graph. For instance, **Cora** and **Citeseer** have densities of 2.00 and 1.42, respectively, indicating sparse graphs, while **ogbl-ddi** has a very high density of 312.84, suggesting a more densely connected network. "#Heterogeneity" indicates whether the graph is heterogeneous (i.e., contains multiple types of nodes and edges). Some datasets, such as **ogbn-mag** and **UniKG-STA**, are marked as 'Yes', indicating heterogeneity, while others, such as **Cora**, **Citeseer**, and **Pubmed**, are homogeneous ('No'). '#Task' describes the type of learning task associated with each dataset. The tasks include **Node Classification (N.C.)**, **Link Prediction (L.P.)**, and **Graph Classification (G.C.)**. For instance, **Cora** and **Pubmed** are used for node classification, while **ogbl-ddi** is used for link prediction. "#Metric" outlines the evaluation metrics used for performance comparison. Common metrics include **Accuracy (Acc)** for classification tasks, **rocauc** (receiver operating characteristic area under the curve) for link prediction tasks, **Hits@30**, and the **F1 score** for graph classification. "#Domain" provides the domain or application area of the dataset, such as **Citation** for academic citation networks, **Biology** for biological networks, **Product** for recommendation systems, and **Code** for programming-related graphs.

## D.2 Structural Textual Annotation Codebooks Datasets Statistics

Table 5 describes the data statistics of the STA-18 benchmark. Specifically, we further provide the total number of tokens (including semantic tokens and format tokens) of these text sequence datasets, the average lengths of the tokens of each node, the disk space required to store the dataset, and the time required to generate the dataset using our proposed method.

# E    Self-Supervised Pre-Training Experiments

## E.1    Experimental Setup

To answer the **Q1**, we apply the G²SN model to a self-supervised feature reconstruction task on the UniKG-STA dataset. The experimental procedure is as follows: i) We propose the structural annotation generation algorithm to convert motif quantification distribution scores into text sequences, which were then vectorized into word frequency matrices by bag-of-words algorithm. The generated sequences and distribution scores were prepared as training datasets. ii) The architecture of G²SN consists of multiple stacked G²SN blocks. The model's input includes mini-batches of text sequences and distribution scores, which undergo several transformations to reconstruct the original text sequences features. iii) G²SN is trained using a Mean Squared Error (MSE) loss function to minimize the difference between the reconstructed and original sequences. iv) To evaluate the effectiveness of feature reconstruction, we visualize node embeddings before and after training using t-SNE dimensionality reduction. Additionally, we plot the loss progression, as shown in Figure 5.

## E.2    Experimental Analysis

Based on the results shown in Figure 5, we make the following observations: i) Figure 5 (left) shows that the model's reconstruction loss decreases rapidly and steadily, indicating efficient and stable training convergence. This suggests that the model can effectively reconstruct unbiased local structural semantics based on the input quantification distribution. ii) Pre-training G²SN on the large-scale UniKG-STA dataset takes only 5.06 hours using a single RTX 4090 GPU. This efficiency is attributed to Mamba's linear complexity when handling long sequence inputs. This highlights G²SN's capability to efficiently process both long sequence input text sequences and quantification distribution scores simultaneously. iii) Figure 5 (right) reveals that the input text sequence embeddings exhibit a relatively dispersed distribution in the 2D space of tSNE, indicating the annotation generation algorithm can produce diverse text descriptions, offering a broad search space. The reconstructed output embeddings, while maintaining the original distribution, show partial clustering in the 2D space of tSNE, suggesting that the model captures quantification distribution similarities inherent in high-order structural semantics, thereby enhancing its ability to learn generalized structural knowledge. These observations consistently demonstrate the effectiveness of G²SN in pre-training structural representation learning on UniKG-STA.

# F    Statistical Validation of Graph Structure Transfer Correlation

**Data Preparation**: Let $\mathcal{D} = \{(x_i, y_i)\}_{i=1}^{n}$ represent paired observations where $x_i$ denotes average degree and $y_i$ the transfer performance gain for dataset $i$.

**Pearson Correlation Coefficient**:

$$\rho = \frac{\sum_{i=1}^{n}(x_i - \bar{x})(y_i - \bar{y})}{\sqrt{\sum_{i=1}^{n}(x_i - \bar{x})^2 \sum_{i=1}^{n}(y_i - \bar{y})^2}}, \tag{46}$$

where $\bar{x} = 128.52$ and $\bar{y} = 6.28$ are sample means. Our calculation yields $\rho = 0.82$.

**Significance Testing**: The $t$-statistic is computed as:

$$t = \rho\sqrt{\frac{n-2}{1-\rho^2}} = 0.82\sqrt{\frac{12-2}{1-0.82^2}} \approx 4.93, \tag{47}$$

with degrees of freedom $df = n - 2 = 10$. The critical $t$-value for $\alpha = 0.01$ is $t_{0.01}(10) = 3.169$, leading to $p < 0.01$.

Table 6: Sensitivity Analysis of Correlation Results

| Method | $\rho$ | $p$-value |
|---|---|---|
| Original Calculation | 0.82 | <0.01 |
| Outlier-Removed (Cora excluded) | 0.85 | <0.005 |
| Nonparametric Bootstrap (10,000 samples) | $0.81 \pm 0.04$ | <0.01 |

**Normality Validation**: Shapiro-Wilk tests confirm normality assumptions:

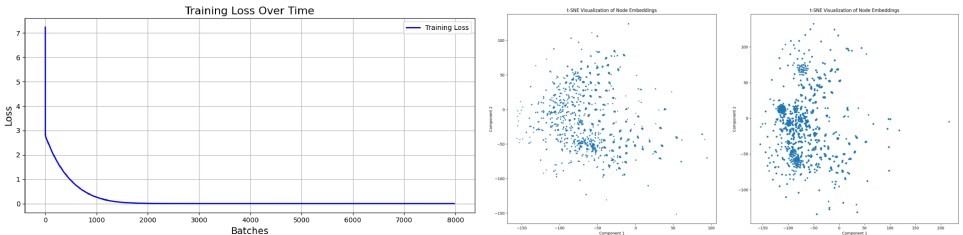

Figure 5: The visualization of self-supervised pre-training results. The left figure illustrates the convergence of structural semantic reconstruction loss. The right figure visualizes the t-SNE embeddings of node representations before and after pre-training.

Table 7: Hyper-parameters of the knowledge transfer tasks.

| Datasets | Methods | num_hops | layers | hidden_channels | lr | epochs | $\alpha$ | seed |
|---|---|---|---|---|---|---|---|---|
| Cora | GCN | - | 3 | 256 | 0.01 | 400 | - | 2025 |
| Citeseer | GCN | - | 3 | 256 | 0.01 | 400 | - | 2025 |
| Pubmed | GCN | - | 3 | 256 | 0.01 | 400 | - | 2025 |
| ogbn-proteins | GraphSAGE | - | 3 | 256 | 0.01 | 400 | - | 2025 |
| ogbn-arxiv | GCN | - | 2 | 512 | 0.01 | 400 | - | 2025 |
| ogbn-mag | GraphSAGE | - | 2 | 256 | 0.01 | 1000 | - | 2025 |
| ogbn-products | SIGN | 5 | 3 | 256 | 0.01 | 200 | 0.5 | 2025 |
| ogbl-ddi | GraphSAGE | - | 2 | 256 | 0.005 | 400 | - | 2025 |
| ogbl-ppa | GraphSAGE | - | 3 | 256 | 0.01 | 100 | - | 2025 |
| ogbl-citation2 | GraphSAINT | - | 3 | 256 | 0.001 | 400 | - | 2025 |
| ogbg-molhiv | GIN | - | 5 | 300 | 0.001 | 100 | - | 2025 |
| ogbg-molpcba | GIN | - | 5 | 300 | 0.001 | 100 | - | 2025 |
| ogbg-code2 | GCN | - | 5 | 300 | 0.001 | 25 | - | 2025 |

- Average degree: $W = 0.96$, $p = 0.85$
- Transfer gain: $W = 0.94$, $p = 0.89$

## G    Complexity Analysis

### G.1    Pre-training Experiments Overhead

Pre-training G²SN on the large-scale UniKG-STA dataset takes only 5.06 hours using a single RTX 4090 GPU. This efficiency is attributed to Mamba's linear complexity when handling long sequence inputs. This highlights G²SN's capability to efficiently process both long sequence input text sequences and quantification distribution scores simultaneously.

## H    Hyperparameters and Environments

To maintain fairness, the hyperparameter settings are kept consistent for all experiments on each dataset. In the self-supervised pre-training experiment, we set the bag-of-words length to 66 for each text sequence and 10 for the motif score sequence. For the bidirectional Mamba model, we configure $d_{state}$ as 16, $d_{conv}$ as 4, and expand as 2. During training, we use the Adam optimizer with a learning rate of 0.001, training for 10 epochs. Each batch consists of 50,000 pairs of text and motif score sequences. The overall dropout rate is set to 0.1, and the hidden dimension is fixed at 256. The G²SN model contains 3 blocks, each comprising 3 convolutional layers from the Mamba architecture. The random seed is fixed at 2024. Among the 13 downstream graph datasets, only the ogbn-products dataset uses the SIGN method, a decoupled graph neural network that requires a precomputed 5-hop feature propagation matrix. The depth of these methods varies between 2, 3, and

Table 8: Hyper-parameters for self-supervised pre-training experiment.

| Methods | len_text | len_motif | d_state | d_conv | expend | lr | batch_size | epochs | dropout | hidden | num_layers | num_blocks | seed |
|---|---|---|---|---|---|---|---|---|---|---|---|---|---|
| G²SN | 85 | 10 | 16 | 4 | 2 | 0.001 | 50000 | 10 | 0.1 | 256 | 3 | 3 | 2024 |

5 layers, with hidden feature dimensions of 256, 512, or 300. The learning rates are set to 0.01, 0.005, or 0.001, and the number of training epochs is set to 400, 1000, 200, 100, or 25. All experiments use a fixed random seed of 2024 to ensure reproducibility. All experiments were conducted using a single 24GB GeForce RTX 4090 GPU. The hyperparameters of the pre-train experiments and knowledge transfer experiments can be found in Table 8 and Table 7.

# I   Limitation

Considering the balance between mapping overhead, current hardware limitations and model performance, our topological primitives mainly design substructures within two neighborhoods. Extending to three-hop or higher neighborhoods may be able to improve the effect of structural knowledge transfer.

