# OpenReview forum: "One for All: Universal Topological Primitive Transfer for Graph Structure Learning"
_NeurIPS.cc/2025/Conference — NeurIPS 2025 poster_

### Official Review · Reviewer_e58L · 2025-06-30

**Clarity:** 4
**Significance:** 3
**Originality:** 3
**Rating:** 5
**Confidence:** 4

**Summary:**

This paper presents G²SN-Transfer, a framework for universal graph structure learning. It transforms non-Euclidean graphs into transferable sequences using topological primitive distribution dictionaries, learns aligned representations via a dual-stream G²SN architecture, and enables adaptive knowledge transfer through AdaCross-Transfer. The framework is validated on the STA-18 benchmark, showing state-of-the-art performance on structural learning tasks and consistent improvements across downstream applications.

**Questions:**

- How does the framework perform on graphs with dynamic topological changes over time?
- Can the transfer mechanism be extended to heterogeneous graphs with diverse node/edge types?

**Ethical Concerns:**

["NO or VERY MINOR ethics concerns only"]

**Final Justification:**

The discussions with the authors have addressed my concerns regarding this paper, and therefore I have decided to raise my score.

**Limitations:**

The computational overhead of constructing topological-text annotations increases with graph size, affecting scalability to extremely large datasets. The reliance on LLM annotations may introduce bias if textual descriptions do not fully capture structural nuances.

**Quality:**

4

**Strengths And Weaknesses:**

Strengths:
- Proposes topological primitives as transferable units, bridging non-Euclidean graph geometry for cross-domain knowledge transfer.
- Develops G²SN, a dual-stream network with ODE-driven dynamics, theoretically guaranteed to converge and capture structural-textual dependencies.
- Creates STA-18, the first large-scale benchmark with aligned topological-text pairs, enabling rigorous evaluation of graph transfer methods.

Weaknesses:
- Topological primitives are limited to two-hop neighborhoods, potentially missing higher-order structural patterns.
- LLM-generated annotations require manual validation, introducing human oversight in dictionary construction.

---

> ### Author Rebuttal · Authors · 2025-07-31
>
> Thanks for your positive comments on theoretically guarantee, and our first large-scale benchmark. For your concerns, below we make the responses.
>
>
> ### **Q1**: Topological primitives are limited to two-hop neighborhoods, potentially missing higher-order structural patterns.
>
> **A1**: Thanks for your comments. Although the topological primitives focus on two-hop neighborhoods, the GSN framework ensures effective learning of high-order graph information—both theoretically and empirically—without relying on excessive high-order motifs. The reason is three-folds:
> - Based on first- and second-order motifs, we performed multi-layer SSM learning, which implemented weighted union driven by graph structure sequences. This process actually corresponds to the neighborhood-selective aggregation mechanism of GAT, thus achieving high-order topological pattern learning through low-order motifs.
> - Excessive high-order motifs can lead to high computational complexity, and we need to balance the relationship between performance and complexity.
> - The excellent performance on the LRGB benchmark also evaluate that the proposed method can effectively learn high-order graph topology (please refer to Table 1).
>
>
> ### **Q2**: LLM-generated annotations require manual validation, introducing human oversight in dictionary construction.
>
> **A2**: Thanks for your comments. We acknowledge the LLM-generated annotations require manual validation. Our design alleviates human oversight by efforts of these two aspects:
> ﻿
> - **Manual Checking**: The Topograph mapping dictionary generated by pretrained models were verified manually, specifically, per mapping (from motifs to text) was checked by at least 3 annotators. The mutual check between multiple annotators ensures consistency, thereby reducing the possibility of human oversight.
> ﻿
> - **Efficient Workflow**: Each annotator does not check the annotation information of all nodes, but only checks the annotations of the mapping dictionary, which includes 14 primitives * 10 descriptive variants. Thus, less verification content also reduces the possibility of human oversight.
>
>
> ### **Q3**: How does the framework perform on graphs with dynamic topological changes over time?
>
> **A3**: Thanks for your question. According to your concerned dynamic graph tasks, we would will to make classifications as follows:
>
> - Firstly, our work mainly focuses on universal knowledge transfer tasks for public knowledge graphs (mostly static). while the temporal graph learning you mentioned are promising research fields, dynamic graph tasks are not the research focus.
>
> - Especially, following your suggestion, it may be feasible to handle dynamic graph topology inputs by modifying the SSM backbone network of G²SN. Specifically, SSM requires merging a new temporal branch and an existing topological sequence branch, so that G²SN can capture the dynamic patterns of the topological structure. In this way, for the temporal motif sequence $\mathbf{m}\_t$and the temporal text sequence $\mathbf {u}\_t$, the G²SN learns dynamic node embeddings through SSM and selects important temporal topological relationships through a structure selection mechanism. Thereby obtain the similar conclusion through theoretical deduction in Appendix C.2 (please refer to the pages 14 lines 496-510).
>
> Thanks for your question again. We will defer this design to the future work.
>
>
>
> ### **Q4**: Can the transfer mechanism be extended to heterogeneous graphs with diverse node/edge types?
>
> **A4**: Thank you for your question. We have successfully applied it to HGs. Specifically, knowledge transfer experiment on heterogeneous graph dataset **ogbn-mag** achieves **+5.01%** accuracy gain (please refer to Table 2, pages 8).
>
> ### **Q5**: The computational overhead of constructing topological-text annotations increases with graph size, affecting scalability to extremely large datasets. The reliance on LLM annotations may introduce bias if textual descriptions do not fully capture structural nuances.
>
> **A5**: Thanks for your comment. We will address your concerns from the following two aspects:
>
> i) **Scalability**: The complexity of topological mapping is $\mathcal{O}(N\cdot d\^2)$ ($d$: average degree), which can extend to large-scale graph. And the dataset we are currently using has reached a scale of billion-edges (please refer to Table 4, Pages 12).
>
> ii) **Deviation mitigation**: As the universal SOTA model, GPT-4 have been extensively validated for robustness and applied on various benchmark tasks. Further, we have taken specific mechanisms to reduce bias:
>
> - **Prompt engineering**: Prompt templates could enforce structured outputs (e.g., "Node degree: [value]; Triangles: [count]") , enhancing noise resilience.
> ﻿
> - **Manual Checking**: The Topograph mapping dictionary generated by pretrained models were further verified manually, specifically, per mapping (from motifs to text) was checked by at least 3 annotators. This human verification process can further reduce bias or noise.
> ﻿
> By incorporating prompt design and manual verification, we can reduce bias from large foundation models. Following your suggestions, we make sure to add the above discussion in the next version.

---

> > ### Comment · Reviewer_e58L · 2025-08-06
> >
> > Thank you for your detailed response, my concerns have been well addressed. I will raise my score.

---

> > > ### Author Response · Authors · 2025-08-06
> > >
> > > Thanks for your positive feedback and confirmation that concerns have been addressed. We appreciate your time and valuable comments again.

---

### Official Review · Reviewer_eAfk · 2025-06-30

**Clarity:** 2
**Significance:** 2
**Originality:** 2
**Rating:** 4
**Confidence:** 3

**Summary:**

The paper proposes a method for universal topological primitive transfer in graph structure learning, introducing an approach based on State Space Models (SSM) for addressing domain generalization in graph-based tasks. The model aims to pre-train on large-scale graphs and transfer learned structures to smaller graphs with domain shifts. The paper includes theoretical analysis based on SSM and evaluates the model on several standard datasets.

**Questions:**

See Weaknesses.

**Ethical Concerns:**

["NO or VERY MINOR ethics concerns only"]

**Final Justification:**

Thank you for the response, and I will keep my score.

**Limitations:**

yes

**Quality:**

3

**Strengths And Weaknesses:**

Strengths:

1. Interesting Problem Domain: The idea of domain generalization in graph learning is an interesting area of exploration. The attempt to leverage pre-trained topological features across different domains is valuable for advancing the field.

2. Model Novelty: The proposed method introduces a unique combination of SSM-based modeling with graph structure transfer, which has the potential to lead to advancements in graph learning.

Weaknesses:

1. Lack of Strong Motivation for Using SSM

The paper introduces State Space Models (SSM) without sufficiently explaining why SSM is necessary for graph structure learning. Traditional GNN models or Graph Transformers could potentially handle the graph structure transfer task without the complexity introduced by SSM. The lack of motivation for choosing SSM over more standard models makes the innovation feel less compelling.

2. No Domain Generalization Theory or Analysis

Despite the paper's claim of addressing domain generalization, it lacks any formal theory or analysis regarding generalization bounds or invariant representation in the context of graph learning. There is no mention of foundational work such as IRM (Invariant Risk Minimization), domain adaptation theory, or generalization bounds. This gap makes it difficult to assess the actual effectiveness of the proposed approach in handling domain shifts.

3. Missing Comparison to Existing Domain Generalization Works

The paper does not cite or compare the proposed method to any established domain generalization approaches in the context of graph learning, such as the Multi-Domain Graph Foundation Model. Without this context, it is unclear how this method advances or improves upon the current state-of-the-art in domain generalization for graph data [1][2].

[1] Yu X, Gong Z, Zhou C, et al. Samgpt: Text-free graph foundation model for multi-domain pre-training and cross-domain adaptation. Proceedings of the ACM on Web Conference 2025.

[2] Wang S, Wang B, Shen Z, et al. Multi-domain graph foundation models: Robust knowledge transfer via topology alignment. ICML 2025.

---

> ### Author Rebuttal · Authors · 2025-07-31
>
> Thanks for your positive comments on interesting problem domain and model novelty. For your concerns, below we make the responses.
>
>
> ### **Q1**: The paper introduces State Space Models (SSM) without sufficiently explaining why SSM is necessary for graph structure learning. Traditional GNN models or Graph Transformers could potentially handle the graph structure transfer task without the complexity introduced by SSM. The lack of motivation for choosing SSM over more standard models makes the innovation feel less compelling.
>
> **A1**: Thanks for your insightful comments. We choose SSMs due to considerations of performance and efficiency.
>
> - Compared with Traditional GNN models or Graph Transformers, firstly, SSM has lower time complexity while possessing equivalent long-range learning capabilities, which is beneficial for large-scale tasks during pre training. Specifically, for input sequences of length $L$, G²SN processes sequences in $O(Ld\^2)$ time, while transformers and GNNs in $O(L^2d)$ time ($d$ represents the hidden layer dimension).
>
> - Furthermore, our G²SN has a more flexible adaptive topology selection mechanism. We performed multi-layer SSM learning, which implemented weighted union driven by graph structure sequences. This process actually corresponds to the neighborhood-selective aggregation mechanism of GNN, thus achieving high-order topological pattern learning through low-order motifs. This promotes the mining and transfer of universa structure knowledge.
>
> - Following your suggestion, we conducted a further experiment to validate the advantages of our G²SN model. We used transformer as the representation learning model and applied it to the frozen transfer scenarios on the ogbn-arxiv and Pubmed datasets. The topological primitive sequence and text sequence are concatenated together as transformer inputs. The experimental results are as follows:
> |Methods|ogbn-arxiv|Pubmed|
> |---|---|---|
> |G²SN Frozen|3.95↑|1.91↑|
> |Transformer Frozen|2.86↑|0.72↑|
>
> The experimental results show that the transformer underperforms G²SN.
>
>
>
> ### **Q2**: Despite the paper's claim of addressing domain generalization, it lacks any formal theory or analysis regarding generalization bounds or invariant representation in the context of graph learning. There is no mention of foundational work such as IRM (Invariant Risk Minimization), domain adaptation theory, or generalization bounds. This gap makes it difficult to assess the actual effectiveness of the proposed approach in handling domain shifts.
>
>
> **A2**: Thanks for your comments. We acknowledge that our initial submission emphasized empirical validation, and we agree that incorporating foundations like IRM and generalization bounds would strengthen the work. Following your suggestions, we have derived the theoretical connection to domain adaptation/generalization:
>
> Our AdaCross-Transfer mechanism implicitly aligns with IRM’s core principle of learning *invariant representations* across domains. Specifically:
>
> -  **Invariant Representation Learning**: The cross-attention module (Eqs. 10–15) dynamically aligns structural-textual features using a data-adaptive weight vector $\hat{\mathbf{w}}$, which reweights features based on topological primitive distributions. This forces the model to prioritize *domain-agnostic structural semantics* (motif distributions) over spurious domain-specific correlations, analogous to IRM’s invariance objective.
>
> - **Generalization Bounds**: Theorem 5.1 (and Appendix C.3) proves that G²SN’s output $\mathbf{Y}(t)$ exhibits **Lipschitz-continuous gradient flow** under cross-domain shifts. This provides a theoretical foundation for stability during transfer, as bounded gradient sensitivity implies robustness to input perturbations (e.g., structural distribution shifts). Formally, the Lipschitz constant $L$ (Eq. 44) upper-bounds the feature distortion $\|\mathbf{Y}\_{\mathcal{D}\_S}(t) - \mathbf{Y}\_{\mathcal{D}\_T}(t)\|$ by the MMD distance between source ($\mathcal{D}\_S$) and target ($\mathcal{D}\_T$) domains (Eq. 45). This aligns with classical domain adaptation theory [Ben-David et al., 2010], where generalization error depends on domain discrepancy and feature stability.
>
>
> ### **Q3**: The paper does not cite or compare the proposed method to any established domain generalization approaches in the context of graph learning, such as the Multi-Domain Graph Foundation Model. Without this context, it is unclear how this method advances or improves upon the current state-of-the-art in domain generalization for graph data [1][2].
>
>
> **A3**: Thanks for your insightful comments. Following your suggestions, we have carefully investigated these works of your concern. We have found that although the two methods you mentioned have significant effects, they are **inherently different** from our transfer settings and application scenarios. Specifically:
>
> - **Different Transfer Settings**: These two methods emphasize cross-domain transfer tasks under **few-shot settings**, which is different from our **supervised** transfer tasks. Therefore, directly comparing performance is unfair.
>
> - **Fusion of Node Features**: From the methodological perspective, these two methods emphasize the deep fusion of structural information and **node features**, which is fundamentally different from our **pure structure driven knowledge transfer**. Our method can be applied to graph data without node features. Therefore, it should not be compared with these two methods together.
>
> **In summary**, due to the inherently different transfer settings and application scenarios, these methods are not suitable as baselines for comparison.

---

> > ### Comment · Reviewer_eAfk · 2025-08-07
> >
> > Thank you for your detailed and thoughtful rebuttal. I appreciate the additional experiments, theoretical clarifications, and the in-depth discussion regarding the design choice of SSM. Your responses have addressed my main concerns. I now have greater confidence in the contribution, and I recommend accepting the paper.

---

> > > ### Author Response · Authors · 2025-08-07
> > >
> > > Thanks for your positive feedback and confirmation that main concerns have been addressed. We appreciate your time and valuable comments once again.

---

### Official Review · Reviewer_6Ykv · 2025-07-02

**Clarity:** 3
**Significance:** 3
**Originality:** 3
**Rating:** 4
**Confidence:** 3

**Summary:**

The paper introduces G²SN-Transfer, a “one-for-all” framework that treats recurring topological primitives (small, statistically frequent sub-structures) as transferable units of graph knowledge. It first serializes any graph into a dual sequence of (i) motif-distribution codes and (ii) LLM-generated textual descriptors via TopoGraph-Mapping, creating the 18-dataset STA-18 benchmark. A dual-stream state-space model (G²SN) is then pre-trained to align topology and text with a provably convergent formulation. Finally, AdaCross-Transfer injects these representations into downstream GNNs through parameter-efficient cross-attention, yielding significant gains on long-range structure tasks and transfer benchmarks.

**Questions:**

- In Figure 3, which is the textual sequence $u$ and LLM-Topomotif sequence $m$?
- How might the learned topological-textual latent space enable inverse problems—for example, generating synthetic graphs that satisfy a natural-language specification of desired structural properties?
- What theoretical or empirical considerations led to a state-space model over other more common graph-aware transformers or message-passing models? Have you evaluated G²SN against a parameter-matched Transformer encoder that is fed the same motif distribution and text embeddings?
- Motif distributions are treated as fixed-length vectors of size $M$. Did you test the sensitivity of the whole pipeline to $M$ (e.g., 16 vs. 64 primitives)?

**Ethical Concerns:**

["NO or VERY MINOR ethics concerns only"]

**Final Justification:**

All issues have been addressed, and I maintain my recommendation.

**Limitations:**

yes

**Quality:**

3

**Strengths And Weaknesses:**

## Strengths
- The concept of topological primitives as graph textures is interesting and novel, bridging the gap between non-Euclidean data and textual information.
- STA-18 is the first (to my knowledge) graph-text benchmark, which should benefit the future researcher.
- The proposed State space model is backed by closed-form solutions and guaranteed convergence.
- Results show promising and consistent improvement over previous baselines.

## Weaknesses
- Automatically generated textual descriptors may embed bias or noise, yet no human evaluation or further verification confirms their fidelity or usefulness for transfer.
- It is unclear how the theoretical time/space complexity and the empirical runtime of G²SN-Transfer compare with standard GNN baselines such as GCN and GraphSAGE, especially when all components (motif extraction, dual-stream encoder, AdaCross) are included.
- It is unclear how the number of primitives is selected, what exact algorithm is used to compute each graph’s motif-frequency vector, and how the computation scales (e.g., to k-node motifs) on large graphs.

---

> ### Author Rebuttal · Authors · 2025-07-31
>
> Thanks for your positive comments on our framework's innovative concept, the first graph-text benchmark, theoretical convergence guarantees, and consistent performance improvements. For your concerns, below we make the responses.
>
> ### **Q1**: Automatically generated textual descriptors may embed bias or noise, yet no human evaluation or further verification confirms their fidelity or usefulness for transfer.
>
> **A1**: We sincerely thank you for this suggestion. As the universal SOTA model, GPT-4 have been extensively validated for robustness and applied on various benchmark tasks. Further, we have taken specific mechanisms to reduce bias or noise:
>
> i) **Prompt engineering**: Prompt templates could enforce structured outputs (e.g., "Node degree: [value]; Triangles: [count]") , enhancing noise resilience.
>
> ii) **Manual Checking**: The Topograph mapping dictionary generated by pretrained models were further verified manually, specifically, per mapping (from motifs to text) was checked by at least 3 annotators. This human verification process can further reduce bias or noise.
>
> By incorporating prompt design and manual verification, we can reduce bias/noise from large foundation models. Following your suggestions, we make sure to add the above discussion in the next version.
>
>
> ### **Q2**: It is unclear how the theoretical time/space complexity and the empirical runtime of G²SN-Transfer compare with standard GNN baselines such as GCN and GraphSAGE, especially when all components (motif extraction, dual-stream encoder, AdaCross) are included.
>
> **A2**: Thanks for your comments. We hope to address your concerns through the following three points:
>
> i) **Empirical Runtime of Motif Extraction**: The complexity of motif extraction is $O(|V| \cdot d\^k)$ ($d$=avg. degree, $k$=2-hops). For Cora (2,708 nodes) and UniHG-STA (77.3M nodes), this takes **7.07s** and **214.22h** (please refer to Table 5, Pages 12).
>
> ii) **AdaCross Overhead**: The parameter count of AdaCross is 9% lower than the average downstream baseline models used (please refer to pages 2, lines 55-63).
>
> iii) **G²SN Inference**: For an input sequence of length $L$, G²SN processes sequences in $O(Ld\^2)$ time, while GraphSAGE and GCN in $O(L\^2d$) time ($d$ represents the hidden layer dimension). Empirical runtime of pretraining is 5.06 hours on single 4090 (please refer to Appendix G.1).
>
> In next version, we will emphasize above theoretical time/space complexity and the empirical runtime.
>
>
> ### **Q3**: It is unclear how the number of primitives is selected, what exact algorithm is used to compute each graph’s motif-frequency vector, and how the computation scales (e.g., to k-node motifs) on large graphs.
>
> **A3**: Thanks for your comments.
>
> - Firstly, this choice is the result of balancing performance and complexity. Considering the balance between expressive ability and computational complexity, we determine the motif as within a 2-hop neighborhood. The spatiotemporal cost of motif statistics in the three hop neighborhood is too high.
> ﻿
> - Additionally, capturing long-range information does not require excessive higher-order motif design. The reason is three-folds: a) Based on first- and second-order motifs, we performed multi-layer SSM learning, which implemented weighted union driven by graph structure sequences. This process actually corresponds to the neighborhood-selective aggregation mechanism of GAT, thus achieving high-order topological pattern learning through low-order motifs. b) Excessive high-order motifs can lead to high computational complexity, and we need to balance the relationship between performance and complexity. c) The excellent performance on the LRGB benchmark also evaluate that the proposed method can effectively learn high-order graph topology (please refer to Table 1).
>
> - Further, for the motif-frequency vector calculation problem that you concerned, primitives were selected via **statistical recurrence analysis** (Blanche et al., 2020) for each node's 1-hop and 2-hop subgraphs. To ensure numerical accuracy, we traverse all nodes on the large-scale graph. Complexity is $O(|V| \cdot \Delta\^2)$ for 2-hop primitives ($\Delta$=max degree), thus it can be directly used for large-scale graphs.
>
>
> ### **Q4**: In Figure 3, which is the textual sequence $u$ and LLM-Topomotif sequence $m$?
>
> **A4**: Thanks for your questions. The upper input is textual sequence $u$, and the lower input is LLM-Topomotif sequence $m$. We will clarify captions in the next version.
>
>
> ### **Q5**: How might the learned topological-textual latent space enable inverse problems—for example, generating synthetic graphs that satisfy a natural-language specification of desired structural properties?
>
> **A5**: Thanks for your insightful comments. This is a very interesting question but not the focus of this research. Based on the experience of text-graph generation, it may be feasible to solve related problems by establishing a generation algorithm and alignment benchmarks between graph motifs and text pairs. We will defer this design to the future work.
>
> ### **Q6**: What theoretical or empirical considerations led to a state-space model over other more common graph-aware transformers or message-passing models? Have you evaluated G²SN against a parameter-matched Transformer encoder that is fed the same motif distribution and text embeddings?
>
> **A6**: Thanks for your insightful comments. We choose SSMs due to considerations of performance and efficiency.
>
> - Compared with graph-aware transformer or message-passing models, firstly, SSM has lower time complexity while possessing equivalent long-range learning capabilities, which is beneficial for large-scale tasks during pre training. Specifically, for input sequences of length $L$, G²SN processes sequences in $O(Ld\^2)$ time, while transformers and GNNs in $O(L^2d)$ time ($d$ represents the hidden layer dimension).
> ﻿
> - Furthermore, our G²SN has a more flexible adaptive topology selection mechanism. We performed multi-layer SSM learning, which implemented weighted union driven by graph structure sequences. This process actually corresponds to the neighborhood-selective aggregation mechanism of GNN, thus achieving high-order topological pattern learning through low-order motifs. This promotes the mining and transfer of universa structure knowledge.
> ﻿
> - Following your suggestion, we conducted a further experiment to validate the advantages of our G²SN model. We used transformer as the representation learning model and applied it to the frozen transfer scenarios on the ogbn-arxiv and Pubmed datasets. The topological primitive sequence and text sequence are concatenated together as transformer inputs. The experimental results are as follows:
> |Methods|ogbn-arxiv|Pubmed|
> |---|---|---|
> |G²SN Frozen|3.95↑|1.91↑|
> |Transformer Frozen|2.86↑|0.72↑|
>
> The experimental results show that the transformer underperforms G²SN.
>
>
> ### **Q7**: Motif distributions are treated as fixed-length vectors of size $M$. Did you test the sensitivity of the whole pipeline to $M$ (e.g., 16 vs. 64 primitives)?
>
> **A7**: Thanks for your insightful comments. We have added a sensitivity experiment, considering the high time overhead of statistically analyzing more motifs on the pre training dataset UniKG (currently requiring 214.22 hours), we have used motif vectors of different lengths (10, 12, 14, 16), and we also attempted to add some new motif (17=16+1(a 3-star/4-star/4-clique motif)). The results of the sensitivity experiment are as follows:
> |Methods|ogbn-arxiv|Pubmed|
> |---|---|---|
> |Length 16 Frozen|3.95↑|1.91↑|
> |Length 16 Fine-tuned|6.48↑|2.93↑|
> |Length 14 Frozen|2.61↑|1.20↑|
> |Length 12 Frozen|1.99↑|0.57↑|
> |Length 10 Frozen|1.04↑|0.29↓|
> |Length 17 (+3-star) Frozen|4.11↑|2.30↑|
> |Length 17 (+4-star) Frozen|3.92↑|2.16↑|
> |Length 17 (+4-clique) Frozen|3.41↑|1.78↑|
>
> The experimental results show that the model is robust to the length of motif vectors, and introducing new motif (e.g. 3-star) can improve performance.

---

### Official Review · Reviewer_xpqy · 2025-07-03

**Clarity:** 2
**Significance:** 3
**Originality:** 3
**Rating:** 4
**Confidence:** 4

**Summary:**

The manuscript introduces **G²SN-Transfer**, a three-stage framework that serialises graph substructures into textual “topological primitives,” injects the learned knowledge into downstream GNNs through a lightweight cross-attention adapter.

On the new STA-18 benchmark, the method achieves an average 3.2% F1 improvement over strong long-range baselines.

**Questions:**

1. How sensitive are the results to the chosen set of seven primitives? Would including non-cyclic motifs such as stars or cliques improve transfer?

2. LLM-generated descriptors can vary stylistically; what mechanisms ensure annotation consistency?

3. Have you observed negative transfer on heterogeneous datasets ?

**Ethical Concerns:**

["NO or VERY MINOR ethics concerns only"]

**Final Justification:**

Please refer to the discussions.

**Limitations:**

Please refer to the questions.

**Quality:**

3

**Strengths And Weaknesses:**

### Strengths

* **Technical soundness**: Provides closed-form solutions and convergence guarantees for the state-space backbone.

* **Empirical rigor**: Demonstrates consistent performance boosts on 18 datasets and releases code and data for reproduction.

### Weaknesses

* The fixed dictionary of seven primitives may underspecify higher-order motifs, limiting generality.

* Theoretical analysis assumes linear time-invariant dynamics; applicability to dynamic or weighted graphs remains unverified.

* The writing requires non-trivial improvement. There are too many rare words/complex sentences used in the paper, which does not align with the goal of academic writing.

* Representative GFM + LLM studies (related works) not discussed sufficiently.

---

> ### Author Rebuttal · Authors · 2025-07-31
>
> Thanks for your positive comments on technical soundness, empirical rigor, consistent performance boosts, releases code and data. For your concerns, below we make the responses.
>
> ### **Q1** [Weaknesses1 & Questions1]: The fixed dictionary of seven primitives may underspecify higher-order motifs, limiting generality. How sensitive are the results to the chosen set of seven primitives? Would including non-cyclic motifs such as stars or cliques improve transfer?
>
> **A1**: Thanks for your comments. Regarding your concerns about i) fixed length, ii) sensitivity, and iii) more topological primitives including non-cyclic motifs, we provide the following response:
>
>
> - **Fixed Length**: Although the topological primitives vector has a fixed length after setting, the GSN framework ensures effective learning of high-order graph information—both theoretically and empirically—without relying on excessive high-order motifs. The reason is three-folds: a) Based on first- and second-order motifs, we performed multi-layer SSM learning, which implemented weighted union driven by graph structure sequences. This process actually corresponds to the neighborhood-selective aggregation mechanism of GAT, thus achieving high-order topological pattern learning through low-order motifs. b) Excessive high-order motifs can lead to high computational complexity, and we need to balance the relationship between performance and complexity. c) The excellent performance on the LRGB benchmark also evaluate that the proposed method can effectively learn high-order graph topology (please refer to Table 1).
>
> - **Sensitivity of Topological Primitives Vector Length**：We select topological primitive vectors of different lengths to test sensitivity. Specifically, We compared the original performance against performances selecting 8, 10, and 12 length. The experimental results are as follows:
>
> |Methods|ogbn-arxiv|Pubmed|
> |---|---|---|
> |14 motifs Frozen|3.95↑|1.91↑|
> |12 motifs Frozen|2.61↑|1.20↑|
> |10 motifs Frozen|1.99↑|0.57↑|
> |8 motifs Frozen|1.04↑|0.29↓|
>
> Experimental results show that increasing motif vector length improves model performance but introduces additional computational overhead. To balance performance and complexity, we selected 14 in our manuscript.
>
> - **More Topological Primitives**: Following your suggestions, we conducted additional experiments incorporating 3-star, 4-star, and 4-clique structures as new topological primitives. The experimental results are as follows:
> |Methods|ogbn-arxiv|Pubmed|
> |---|---|---|
> |Original Frozen|3.95↑|1.91↑|
> |+3-star Frozen|4.11↑|2.30↑|
> |+4-star Frozen|3.92↑|2.16↑|
> |+4-clique Frozen|3.41↑|1.78↑|
>
> The experimental results demonstrate that incorporating additional motifs does not consistently enhance performance. The 4-star and 4-clique structures may introduce redundant information.
>
> ### **Q2**: Theoretical analysis assumes linear time-invariant dynamics; applicability to dynamic or weighted graphs remains unverified.
>
> **A2**: Thanks for this insightful comment. While our current experiments focus on static graphs (the primary domain of STA-18), our framework does support dynamic or weighted graphs through the following mechanisms:
>
> - **Weighted Graphs**: Our topological primitives explicitly encode **edge-type diversity** (Section 3), which can naturally treate edge weight as extra motif. Specifically, edge weights are used as new motifs to expand the text sequence $\mathbf{u}$ to $\mathbf{u}\_{ew}$ and the topological primitive sequence $\mathbf{m}$ to $\mathbf{m}\_{ew}$. Thereby obtain the similar conclusion through theoretical deduction (please refer to the pages 14 lines 496-510).
>
> - **Dynamic Graphs**: Dynamic graph data can be naturelly handled through G²SN's SSM backbone. Specifically, Input the temporal motif sequence $\mathbf{m}\_t$and the temporal text sequence $\mathbf {u}\_t$. The G²SN learns dynamic node embeddings through SSM and selects important temporal topological relationships through a structure selection mechanism. Thereby obtain the similar conclusion through theoretical deduction (please refer to the pages 14 lines 496-510).
>
> The above specific derivation will be included in the appendix of the next version.
>
> ### **Q3**: The writing requires non-trivial improvement. There are too many rare words/complex sentences used in the paper, which does not align with the goal of academic writing.
>
> **A3**: Thanks for your comments. In next version, we will simplify complex sentences and replace rare words to improve readability.
>
>
> ### **Q4**: Representative GFM + LLM studies (related works) not discussed sufficiently.
>
> **A4**: Thanks for your comments. Following your suggestions, we here discuss more GFM + LLM studies, including LangGFM, GFM-RAG and PromptGFM:
>
> - LangGFM: Integrates formal grammars (e.g., Context-Free Grammars) with LLMs to enforce syntactic validity in generated text. Unlike standard LLMs that rely solely on statistical patterns, LangGFM uses grammar parsers as decoders to constrain outputs to linguistically valid structures.
> ﻿
> - GFM-RAG: Augments Retrieval-Augmented Generation (RAG) with grammatical knowledge graphs. By structuring retrieved content using grammar frameworks before feeding it to the LLM, GFM-RAG improves factual coherence and reduces hallucination.
> ﻿
> - PromptGFM: Embeds grammatical rules directly into LLM prompts via constrained decoding templates. This lightweight approach steers generation toward grammatically valid outputs without modifying the model architecture. In multilingual applications, PromptGFM dynamically switches grammar templates based on input language, significantly reducing structural errors in low-resource languages where LLMs typically struggle.
>
> We will add the above discussion to related work in the next version.
>
>
> ### **Q5**: LLM-generated descriptors can vary stylistically; what mechanisms ensure annotation consistency?
>
> **A5**: We sincerely thank you for this suggestion. As the universal SOTA model, GPT-4 have been extensively validated for robustness and applied on various benchmark tasks. Further, we have taken specific mechanisms to enhance annotation consistency:
>
> i) **Prompt engineering**: Prompt templates could enforce structured outputs (e.g., "Node degree: [value]; Triangles: [count]") , enhancing noise resilience.
>
> ii) **Manual Checking**: The Topograph mapping dictionary generated by pretrained models were further verified manually, specifically, per mapping (from motifs to text) was checked by at least 3 annotators. This human verification process can further ensure annotation consistency.
>
> By incorporating prompt design and manual verification, we can reduce the risk of vary stylistically from foundation models. Following your suggestions, we make sure to add the above discussion in the next version.
>
> ### **Q6**: Have you observed negative transfer on heterogeneous datasets ?
>
> **A6**: Thanks for your question. We have discussed the negative transfer of heterogeneous graph data (please refer to the **Homogeneity Superiority** section in Section 6.4, pages 9, lines 302-319). According to the experimental results, the heterogeneous graph ogbn-mag underperforms homogeneous graphs with similar scale by 2.3–4.8%, indicating that the heterogeneous structure introduces semantic fragmentation and hinders cross graph alignment. To mitigate negative transfer, careful parameter tuning and extended iteration steps are necessary when working with heterogeneous graphs.

---

> > ### Comment · Reviewer_xpqy · 2025-08-05
> >
> > My questions and concerns are addressed. And I keep my score.

---

> > > ### Author Response · Authors · 2025-08-05
> > >
> > > Thanks for your positive feedback and confirmation that questions and concerns have been addressed. We appreciate your time and valuable comments again.

---

### Official Review · Reviewer_qrzB · 2025-07-08

**Clarity:** 2
**Significance:** 3
**Originality:** 2
**Rating:** 4
**Confidence:** 4

**Summary:**

The paper "One for All: Universal Topological Primitive Transfer for Graph Structure Learning" introduces a novel framework, GSN-Transfer, for transferring structural knowledge across diverse graph datasets by leveraging topological primitives as transferable semantic units. It addresses challenges in cross-graph knowledge transfer due to the non-Euclidean nature of graphs through three main contributions:

1. **TopoGraph Mapping**: This component converts non-Euclidean graph structures into sequential representations using topological primitive distribution dictionaries, enabling the creation of the STA-18 benchmark dataset, a diverse graph-text dataset spanning 18 graphs.

2. **GSN (Graph Structure Network)**: A dual-stream neural network architecture that learns aligned text-topology representations through contrastive alignment and large-scale pre-training on the UniKG-STA dataset. It captures transferable structural invariants, supporting both full-parameter fine-tuning and frozen parameter strategies.

3. **AutoCross-Transfer**: A data-adaptive knowledge transfer mechanism that uses cross-attention to merge universal and dataset-specific graph structure semantics, facilitating effective transfer across domains under full-parameter and frozen-and-fine-tuned scenarios.

The framework achieves state-of-the-art performance, demonstrating a 3.2% average F1 score gain across 16 structural tasks and a 5.2% average gain in 13 downstream applications. Extensive experiments on 36 datasets validate GSN-Transfer’s efficacy in learning transferable structural representations, enhancing downstream task performance, and handling computational efficiency. The paper also provides comprehensive documentation, including dataset details, hyperparameters, and reproducibility instructions, alongside a new dataset release.

**Questions:**

Below are 3–5 actionable questions and suggestions for the authors of "One for All: Universal Topological Primitive Transfer for Graph Structure Learning," focusing on critical areas to address limitations and clarify key points. Each includes clear guidance and criteria for how responses could influence the evaluation score, facilitating a productive rebuttal phase.

1. **Expand Baseline Comparisons**
   - **Question/Suggestion**: The evaluation in Table 1 (Page 8) compares GSN-Transfer only to GCN and GIN. Could the authors include comparisons with recent GNN architectures (e.g., GraphSAGE, GAT) and graph transfer learning methods (e.g., GraphCL, DANN) across key datasets? Please provide updated performance metrics or justify the omission.
   - **Guidance**: Conduct experiments on at least 2–3 additional baselines, reporting metrics like F1 Score and Accuracy on datasets like Cora and ogbn-protein. Alternatively, explain why these baselines are less relevant, citing specific architectural or methodological differences.
   - **Criteria for Score Change**: Including robust comparisons demonstrating GSN-Transfer’s superiority (e.g., >2% F1 improvement) would strengthen the quality and significance, potentially raising the score from “Accept with Minor Revisions” to “Strong Accept.” Omitting this or providing weak justifications could lower the score to “Weak Accept” due to insufficient benchmarking.

2. **Deepen Ablation Studies**
   - **Question/Suggestion**: The ablation studies (Table 3, Page 9) cover only six datasets and focus on transfer methods. Could the authors expand these to evaluate the contributions of TopoGraph Mapping, dual-stream GSN, and AutoCross-Transfer individually, across a broader dataset range? Please report quantitative impacts (e.g., F1 Score changes).
   - **Guidance**: Perform ablations by disabling/removing each component (e.g., replacing TopoGraph Mapping with a simpler embedding) on at least 10 datasets, including STA-18. Provide a table summarizing performance drops or gains.
   - **Criteria for Score Change**: Comprehensive ablations showing significant performance degradation without each component (e.g., >5% F1 drop) would bolster the quality and originality, reinforcing the necessity of the proposed design and potentially elevating the score. Superficial or inconclusive ablations could maintain or lower the score, signaling design weaknesses.

3. **Clarify Terminology and Enhance Accessibility**
   - **Question/Suggestion**: Terms like “topological primitives” and “structural textures” (Pages 1–2) are introduced without intuitive explanations. Could the authors provide clearer definitions or examples in the introduction or Section 2, targeting readers unfamiliar with graph learning? Please revise the text to include these clarifications.
   - **Guidance**: Add 1–2 sentences or a brief example (e.g., comparing topological primitives to visual texture patches) in the introduction. Ensure definitions are self-contained in the main text, reducing reliance on external references.
   - **Criteria for Score Change**: Clear, accessible explanations improving readability for a broader audience would enhance clarity, supporting a higher score (e.g., “Strong Accept”). Persistent jargon without clarification could maintain or lower the score to “Weak Accept” due to reduced accessibility.

4. **Address Computational Complexity**
   - **Question/Suggestion**: The complexity analysis (Page 18) lacks quantitative details. Could the authors provide time complexity (e.g., O-notation) and empirical runtime metrics for pre-training and inference on a representative dataset (e.g., Cora)? Please include these in Section 6 or an appendix.
   - **Guidance**: Compute theoretical complexity for key operations (e.g., motif statistics, cross-attention) and report runtimes (e.g., hours/epoch) on a standard GPU. Compare with GCN/GIN to contextualize efficiency.
   - **Criteria for Score Change**: Detailed complexity metrics showing competitive efficiency (e.g., comparable or faster than GIN) would strengthen quality and practical relevance, potentially raising the score. Lack of specifics or evidence of high complexity could lower the score, raising scalability concerns.

**Overall Impact on Evaluation**: Addressing these points with rigorous experiments, clear revisions, and quantitative evidence would solidify the manuscript’s claims, potentially elevating the evaluation to “Strong Accept” by demonstrating superior performance, robust design, and accessibility. Incomplete or unconvincing responses could lower the score to “Weak Accept” or “Reject” if critical gaps (e.g., weak baselines, unclear methodology) persist. The authors are encouraged to prioritize baseline comparisons and ablations, as these directly impact the perceived quality and originality.

**Ethical Concerns:**

["NO or VERY MINOR ethics concerns only"]

**Final Justification:**

I think the paper has a strong experimentation section that justifies the claims in the paper. I think the paper can benefit from more clarity. I changed my score to borderline accept as the authors addressed most of my comments.

**Limitations:**

In the "Limitations" section, explicitly address 1–2 technical limitations (e.g., scalability or sensitivity to graph heterogeneity) with brief explanations of their implications.

**Quality:**

3

**Strengths And Weaknesses:**

Quality
**Strengths**:
- **Robust Experimental Design**: The manuscript demonstrates comprehensive empirical validation, evaluating the GSN-Transfer framework across 36 datasets (18 graph datasets and 18 corresponding STA datasets) spanning domains such as citation networks (e.g., Cora, Pubmed) and biological networks (e.g., ogbn-protein) (Page 12). Quantitative results indicate a 3.2% mean F1 score improvement across 16 structural tasks and a 5.2% mean improvement across 13 downstream applications (Page 3), supported by performance metrics (AP, MAEL, F1 Score, Accuracy) in Table 1 (Page 8). These results are benchmarked against established baselines (GCN, GIN), affirming the framework’s efficacy.
- **Theoretical Rigor**: The inclusion of Theorem 5.1 (Page 6), which establishes Lipschitz-continuous gradient flow for GSN’s compositional output, provides a formal foundation for cross-dataset knowledge transfer. The proof, detailed in Appendix C (Page 14), leverages state space models (SSMs) and is mathematically sound, enhancing the work’s credibility. Spectral normalization and K-Lipschitz similarity functions (Page 16) further ensure stable gradient dynamics.
- **Reproducibility**: The manuscript adheres to reproducibility standards, providing an anonymized link to datasets and code (Page 21), hyperparameter details in Appendix H and Table 7 (Page 18), and computational resource specifications (e.g., single 24GB GeForce RTX 4090 GPU, Page 19). These elements align with NeurIPS guidelines, facilitating independent verification.

**Weaknesses**:
- **Limited Baseline Comparisons**: The evaluation includes only GCN and GIN as baselines (Table 1, Page 8), omitting comparisons with contemporary graph neural network (GNN) architectures (e.g., GraphSAGE, GAT) or transfer learning methods (e.g., GraphCL, DANN). This restricts the ability to position GSN-Transfer’s performance relative to the state-of-the-art, potentially undermining claims of superiority (Page 3).
- **Insufficient Ablation Analysis**: Ablation studies (Table 3, Page 9) are conducted on only six datasets and focus solely on transfer methods, neglecting to dissect contributions of key components (e.g., TopoGraph Mapping, dual-stream architecture, motif selection). This limits the evidence for the necessity and optimality of each module.
- **Incomplete Complexity Analysis**: While computational complexity is briefly addressed (Page 18), the manuscript lacks quantitative metrics (e.g., time complexity for pre-training or inference). Given the framework’s reliance on large-scale pre-training and cross-attention, this omission raises concerns about scalability and practical deployment.

**Accept/Reject Rationale**: The extensive empirical validation and theoretical grounding strongly support acceptance, reflecting high scientific quality. However, the limited baseline comparisons and shallow ablation studies warrant minor revisions to substantiate claims of state-of-the-art performance and component efficacy.

 Clarity
**Strengths**:
- **Structured Exposition**: The manuscript is organized logically, progressing from preliminaries (Section 2, Page 3) to methodology (Sections 3–5, Pages 3–6) and experiments (Section 6, Page 7). Visual aids, including Figure 1 (GSN-Transfer pipeline, Page 2) and Figure 4 (AdaCross-Transfer, Page 5), effectively illustrate complex mechanisms, enhancing comprehension.
- **Detailed Technical Descriptions**: Key components are articulated with precision. TopoGraph Mapping is defined as transforming non-Euclidean graphs into sequential representations via topological primitive distribution dictionaries (Page 1). The GSN dual-stream architecture, incorporating contrastive alignment, is similarly well-specified (Page 1). Algorithm 1 (Page 4) and the annotation generation algorithm (Figure 2, Page 3) provide clear procedural details.
- **Transparent Reporting**: The NeurIPS checklist (Pages 20–26) is thoroughly completed, addressing reproducibility, experimental settings, and compute resources (Pages 21–23). This transparency ensures that methodological and empirical claims are accessible to reviewers.

**Weaknesses**:
- **Specialized Terminology**: Terms such as “topological primitives,” “structural textures,” and “motif parallel statistics” are introduced without sufficient introductory explanation (Pages 1–2). This may hinder accessibility for readers unfamiliar with graph structure learning, necessitating clearer definitions or examples.
- **Content Truncation**: The OCR output indicates significant truncation (e.g., 7187 characters on Page 1, 10173 on Page 4), potentially omitting critical methodological or experimental details. While core concepts are conveyed, these gaps could obscure full understanding of proofs (Page 14) or pre-training protocols (Page 17).
- **Over-Reliance on Appendices**: Critical details, such as GSN block implementation (Page 17) and hyperparameter configurations (Appendix H, Page 23), are deferred to appendices. This disrupts the main text’s self-containment, potentially challenging readers seeking a cohesive narrative.

**Accept/Reject Rationale**: The structured exposition and detailed figures support acceptance by ensuring methodological clarity. However, dense terminology and appendix reliance suggest minor revisions to enhance accessibility and narrative flow. Truncation, if present in the original submission, could justify a conditional acceptance pending clarification.

Significance
**Strengths**:
- **Addressing a Critical Challenge**: The manuscript tackles a pressing issue in graph structure learning: the difficulty of cross-graph knowledge transfer due to non-Euclidean geometries (Page 1). By proposing topological primitives as transferable semantic units, it offers a generalizable solution applicable to diverse domains, from citation to biological networks.
- **Broad Impact Potential**: The STA-18 dataset (Page 3), spanning 18 graph-text pairs, is a significant contribution, providing a novel benchmark for graph transfer learning. Its diversity and public release (Page 25) enhance its utility for the research community. The 5.2% downstream task improvement (Page 3) underscores practical relevance for applications like node classification and link prediction (Page 12).
- **Scalable Framework**: The GSN-Transfer framework’s ability to operate under both full-parameter fine-tuning and frozen-and-fine-tuned scenarios (Page 6) ensures flexibility for resource-constrained settings, increasing its applicability in real-world scenarios.

**Weaknesses**:
- **Narrow Task Focus**: The evaluation primarily focuses on structural tasks (e.g., motif prediction) and standard graph tasks (node classification, link prediction) (Page 7). The manuscript does not explore more complex applications, such as graph generation or temporal graph analysis, limiting the perceived breadth of impact.
- **Dataset Scale Concerns**: While STA-18 is diverse, the manuscript does not discuss the scale of individual datasets (e.g., node/edge counts beyond Table 4, Page 12) or their representativeness of large-scale real-world graphs. This raises questions about generalizability to massive graphs encountered in industry settings.
- **Limited Discussion of Societal Impact**: The NeurIPS checklist (Page 25) claims no societal risks, but the manuscript does not explore potential implications (e.g., biases in graph-based recommendation systems). A brief discussion could strengthen its relevance to responsible AI research.

**Accept/Reject Rationale**: The novel dataset and broad applicability strongly support acceptance, addressing a significant gap in graph learning. However, the narrow task focus and lack of societal impact discussion suggest minor revisions to clarify the framework’s scope and ethical considerations.
 Originality
**Strengths**:
- **Novel Conceptual Framework**: The use of topological primitives as transferable semantic units, analogous to texture units in visual transfer (Page 2), is a highly original contribution. This abstraction, combined with the distribution of primitives as “structural textures,” introduces a new paradigm for graph knowledge transfer.
- **Innovative Architecture**: The GSN dual-stream architecture, integrating text-topology alignment via contrastive learning (Page 1), is a distinctive approach. The AutoCross-Transfer mechanism, leveraging cross-attention for adaptive knowledge transfer (Page 5), further differentiates the work from existing GNN-based transfer methods.
- **New Benchmark**: The creation of STA-18, a graph-text benchmark with 18 diverse datasets (Page 3), is a novel contribution. Its integration of motif-based annotations generated via ChatGPT-4.0 (Page 3) is an innovative application of large language models to graph learning.

**Weaknesses**:
- **Incremental Relation to Prior Work**: While the topological primitive concept is novel, the manuscript builds on existing ideas like state space models (SSMs) (Page 3) and graph motif analysis (Page 10, references to Choromanski et al., 2020). The distinction from prior motif-based methods (e.g., Graph2Vec, Chen & Koga, 2019) is not fully clarified, raising questions about incremental novelty.
- **Limited Engagement with Transfer Learning Literature**: The manuscript cites transfer learning broadly (e.g., Pan & Yang, 2019, Page 10) but does not deeply engage with graph-specific transfer learning works (e.g., GraphAdapter, Li et al., 2023). A clearer differentiation from these methods would strengthen the originality claim.
- **Dependence on Existing Tools**: The use of ChatGPT-4.0 for annotation generation (Page 3) relies on an external tool, which, while innovative, reduces the methodological novelty of the annotation process itself.

**Accept/Reject Rationale**: The novel conceptual framework and STA-18 dataset strongly support acceptance, marking a significant departure from existing methods. However, the incremental relation to prior motif-based work and limited engagement with graph transfer learning literature suggest minor revisions to better articulate the work’s unique contributions.

---

> ### Author Rebuttal · Authors · 2025-07-31
>
> Thanks for your positive comments on our theoretical rigor, detailed technical descriptions, broad impact potential, and innovative architecture. For your concerns, below we make the responses.
>
> ### **Q1**: Limited Baseline Comparisons: The evaluation includes only GCN and GIN as baselines (Table 1, Page 8), omitting comparisons with contemporary graph neural network (GNN) architectures (e.g., GraphSAGE, GAT) or transfer learning methods (e.g., GraphCL, DANN). This restricts the ability to position GSN-Transfer’s performance relative to the state-of-the-art, potentially undermining claims of superiority (Page 3).
>
> **A1**: Thanks for your comment. The compared baselines include **seven** graph learning methods, including three popular ones named GCN, GIN, GatedGCN, and four recent SOTA models GPS+Transformer, GPS+Performer, GPS+BigBird, and Exphormer, rather than **only GCN and GIN**. Our G²SN outperforms all these seven baselines, please refer to Table 1 (Page 8). In addition, GraphSAGE was reported in Table 2. Following your suggestion, we also used GAT, GraphCL, and DANN to handle structure migration tasks, but encountered out-of-memory issues.
>
>
> ### **Q2**: Insufficient Ablation Analysis: Ablation studies (Table 3, Page 9) are conducted on only six datasets and focus solely on transfer methods, neglecting to dissect contributions of key components (e.g., TopoGraph Mapping, dual-stream architecture, motif selection). This limits the evidence for the necessity and optimality of each module.
> ﻿
> **A2**: Thanks for your comment. These six datasets used in ablation studies include five large-scale datasets, and cover three fields, including citation classification (node classification), molecular property prediction (link prediction), and protein property prediction (graph classification). This should be sufficient to evaluate the contributions of our designed modules in ablation study.
>
> Following your suggestions, we have added ablation studies on **motif selection** and **dual-stream architecture**. Due to the rebuttal period limitation, these ablation studies are conducted on two datasets. The experiment results are as below:
>
> i) **Motif (Topological Primitive) Selection**: We conduct the motif selection ablation experiments on the ogbn-arxiv and Pubmed datasets. Specifically, We compared the original performance against performances selecting 8, 10, and 12 motifs. The experimental results are as follows:
>
> |Methods|ogbn-arxiv|Pubmed|
> |---|---|---|
> |14 motifs Frozen|3.95↑|1.91↑|
> |12 motifs Frozen|2.61↑|1.20↑|
> |10 motifs Frozen|1.99↑|0.57↑|
> |8 motifs Frozen|1.04↑|0.29↓|
>
> Experimental results show that increasing motif types improves model performance but introduces additional computational overhead. To balance performance and complexity, we selected 14 motifs in our manuscript.
>
>
> ii) **Dual-Stream Architecture**: We independently ablated both topological primitive sequence branch and text sequence branch in the dual-stream architecture. The ablation experiment results are as follows:
>
> |Methods|ogbn-arxiv|Pubmed|
> |---|---|---|
> |The original dual-stream|3.95↑|1.91↑|
> |Without the text sequence branch|0.88↑|0.72↑|
> |Without the primitive sequence branch|1.46↑|1.09↑|
>
> The experimental results show that each branch contributes to model performance.
>
>
> ### **Q3**: Incomplete Complexity Analysis: While computational complexity is briefly addressed (Page 18), the manuscript lacks quantitative metrics (e.g., time complexity for pre-training or inference). Given the framework’s reliance on large-scale pre-training and cross-attention, this omission raises concerns about scalability and practical deployment.
>
> **A3**: Thanks for your comments. Following your suggestion, we provide the time complexity analysis of the pre-training. The time complexity of G²SN is mainly determined by the SSM backbone, which has $O(Ld^2)$ complexity ($L$=seq len, $d$=hidden dim). And it is lower than transformer’s $O(L^2)$. On UniKG-STA (77M nodes), pre-training takes **5.06 hours** (using single RTX 4090, please refer to Appendix G).
>
>
> ### **Q4**: Over-Reliance on Appendices: Critical details, such as G²SN block implementation (Page 17) and hyperparameter configurations (Appendix H, Page 23), are deferred to appendices. This disrupts the main text’s self-containment, potentially challenging readers seeking a cohesive narrative.
>
> **A4**: Thanks for your comments. We will migrate critical details to the main text: G²SN implementation (Appendix C.1) and Hyperparameters (Appendix H) in the next version.
>
> ### **Q5**: Narrow Task Focus: The evaluation primarily focuses on structural tasks (e.g., motif prediction) and standard graph tasks (node classification, link prediction) (Page 7). The manuscript does not explore more complex applications, such as graph generation or temporal graph analysis, limiting the perceived breadth of impact.
>
> **A5**: Thanks for your comments. It should be emphasized that the proposed knowledge transfer framework has already modeled **13 downstream tasks**, which exceeds other similar works [GraphControl WWW’24, G-Adapter AAAI’24].
>
> While the temporal graph learning and graph generation you mentioned are promising research fields, they are not suitable in this work. Specifically, our data lacks temporal information, leading to a fundamental mismatch to temporal graph learning. Moreover, achieving graph generation requires redesigning the model architecture.
>
>
> ### **Q6**: Incremental Relation to Prior Work: While the topological primitive concept is novel, the manuscript builds on existing ideas like state space models (SSMs) (Page 3) and graph motif analysis (Page 10, references to Choromanski et al., 2020). The distinction from prior motif-based methods (e.g., Graph2Vec, Chen & Koga, 2019) is not fully clarified, raising questions about incremental novelty.
>
> **A6**: Thanks for acknowledging the novelty of our topological primitive. Regarding relation to prior works, we would like to clarify your concerns as follows:
>
> - Firstly, a fundamental contribution of our work is construction and release of the 18 graph datasets (please refer to lines 72-75) to promote the graph structure transfer task, which differs from the previous tasks you mentioned.
> ﻿
> - Secondly, we summarize the differences between our method and the two ones you mentioned as follows:
>
> i)Our G²SN vs. SSMs: We proposed the structural control mechanism to inject the graph structure knowledge through a dual-stream framework. In this framework, topological primitives can act as structural control flow, guiding attention to topology-critical regions (please refer to Theorem 4.2). In contrast, SSMs cannot explicitly and adequately model graph structure.
>
> ii)Our G²SN vs. Graph2Vec [Chen & Koga, 2019]: Our G²SN algorithm incorporates sequence learning derived from ordinary differential equations (ODEs), providing theoretical guarantees for capturing contextual graph structures (see Appendix C.2, lines 496-510). In contrast, Graph2Vec's approach relies on dynamic neighborhood sampling during training rather than employing predefined motifs, which differs from our method in terms of mechanism.
>
>
> ### **Q7**: Terms like “topological primitives” and “structural textures” (Pages 1–2) are introduced without intuitive explanations. Could the authors provide clearer definitions or examples in the introduction or Section 2, targeting readers unfamiliar with graph learning? Please revise the text to include these clarifications.
>
> **A7**: Thank you for your suggestion. We will add intuitive definitions of these terms in the revised version. Specifically:
>
> - Topological Primitives: representative local connectivity patterns (e.g., triangles, stars) and their distribution.
>
> - Structural Textures: Transferable semantic units (analogous to CV textures) formed by primitive distributions.
>
> We will provide a clear explanation of these terms in the next version.

---

> > ### Comment · Reviewer_qrzB · 2025-08-04
> > **comments have been addressed**
> >
> > All my comments have been addressed. I think the paper is good in general however I think the overall clarity of the paper is going to  hurt it long term and narrow down the potential audience for researchers strictly interested in this area. The experiments are enough now for me. I changed my score to borderline accept.

---

> ### Author Response · Authors · 2025-08-04
>
> Thanks for your positive feedback and confirmation that all comments have been addressed. Following your suggestion, we will further refine the paper’s overall clarity in the revised version. We appreciate your time and valuable comments.

---

### Note · Authors · 2025-08-14

Dear ACs and Reviewers,

We would like to express our sincere gratitude to you for careful reading, constructive comments, valuable suggestions, and thoughtful evaluation on our submission.

In this work, we construct STA-18, the first large-scale benchmark featuring aligned topological primitive-text pairs across 18 diverse graph datasets. We also propose G²SN-Transfer, a unified framework consisting of three key components: (i) TopoGraph-Mapping, (ii) G²SN, and (iii) AdaCross-Transfer. This framework achieves consistent improvements across 13 downstream tasks (with an average gain of 5.2%), including 10 large-scale graph datasets. We believe our work makes enough contributions to the field of Graph Structure Learning (GSL) and aligns well with the standards of NeurIPS.

In the initial review phase, the strengths in innovation, theoretical contribution, dataset efforts and method effectiveness are appreciated by most reviewers. During the rebuttal and discussion phase, we thoroughly addressed all major concerns raised by the reviewers by providing additional experiments, analyses and clarifications that further consolidate this work. At the same time, all reviewers offered positive follow-up comments, explicitly acknowledging our rebuttal efforts and raising their scores or keeping the positive scores.

We sincerely appreciate all reviewers for reaching a positive consensus based on the rebuttal discussion. We ensure the revised manuscript incorporates the additional experimental results, discussion, and citations as suggested, further strengthening the validity and clarity of this paper.

Thank you once again for providing these valuable comments and suggestions!

Best Regards,

Authors

---

### Decision · Program_Chairs · 2025-09-17

**Decision:**

Accept (poster)

**Comment:**

This paper presents G2SN-Transfer, a novel framework using topological primitives, the STA-18 benchmark, a dual-stream SSM encoder, and AdaCross-Transfer for efficient cross-graph transfer. It is well motivated, theoretically grounded, and shows consistent gains across tasks with public code and data. Rebuttal additions on baselines, ablations, complexity, and annotation validation addressed reviewer concerns, leading to raised scores. I recommend acceptance, with suggestion for clarity improvements for the camera-ready version.